# 📊 TSRBENCH: A Comprehensive Multi-task Multi-modal Time Series Reasoning Benchmark for Generalist Models

**Fangxu Yu** [1]  **Xingang Guo** [2]  **Lingzhi Yuan** [1]  **Haoqiang Kang** [3]  **Hongyu Zhao** [1]
**Lianhui Qin** [3]  **Furong Huang** [1]  **Bin Hu** [2]  **Tianyi Zhou** [4]

## Abstract

Time series are ubiquitous in real-world scenarios and crucial for applications ranging from energy management to traffic control. Consequently, the ability to reason over time series is a fundamental skill for generalist models to solve complex problems. However, current benchmarks for generalist models largely overlook this dimension. To bridge this gap, we introduce TSRBENCH, a comprehensive multi-modal benchmark designed to stress-test the full spectrum of time series reasoning capabilities. TSRBENCH features: i) a diverse set of 4125 problems from 14 domains, and is categorized into 4 major dimensions: Perception, Reasoning, Prediction, and Decision-Making. ii) 15 tasks from the 4 dimensions evaluating essential reasoning capabilities (e.g., numerical reasoning). Through extensive experiments, we evaluate over 30 leading proprietary and open-source LLMs, VLMs, and TSLLMs within TSRBENCH. Our findings reveal that: i) scaling laws hold for perception and reasoning but break down for prediction; ii) strong reasoning does not guarantee accurate context-aware forecasting, indicating a decoupling between semantic understanding and numerical prediction; and iii) despite the complementary nature of textual and visual forms of time series as inputs, current multimodal models fail to effectively fuse them for reciprocal performance gains. TSRBENCH provides a standardized evaluation platform that not only highlights existing challenges but also offers valuable insights to advance generalist models. Our code and dataset are available at https://tsrbench.github.io/.

[1]University of Maryland, College Park [2]University of Illinois at Urbana–Champaign [3]University of California, San Diego [4]Mohamed bin Zayed University of Artificial Intelligence. Correspondence to: Tianyi Zhou <tianyi.zhou@mbzuai.ac.ae>.

*Proceedings of the 43rd International Conference on Machine Learning*, Seoul, South Korea. PMLR 306, 2026. Copyright 2026 by the author(s).

## 1. Introduction

Time series pervade real-world environments and underpin decision-making across high-stakes domains, including finance (Dong et al., 2024), healthcare (Morid et al., 2023), and industrial systems (Yan et al., 2024). Since a substantial portion of real-world information is inherently temporal, reasoning on time series becomes a core capability for building generalist models that can reliably solve practical problems. Equipping models with such reasoning abilities enables automated systems to interpret temporal signals in context, supporting downstream applications such as education (Mao et al., 2024), clinical management (Matowe et al., 2003), disaster forecasting (Hakim et al., 2024), and scientific discovery (Yu et al., 2025c).

Given the critical importance of time series reasoning, there is a pressing need for standardized and automated evaluation frameworks that enable comprehensive assessment and comparison. However, existing work remains largely anchored in traditional time series analysis, which adopts a reductive view by treating time series as isolated numerical sequences—thereby stripping away the causal structure and semantic context essential for real-world problem-solving. Recent benchmarks have begun to integrate context (Williams et al., 2024; Liu et al., 2024; Cai et al., 2024a; Kong et al., 2025; Wu et al., 2025), yet they predominantly target surface-level pattern understanding, which is insufficient for complex problem-solving. Other initiatives that attempt to probe reasoning capabilities (Chen et al., 2025b; Wang et al., 2025c; Guan et al., 2025) often remain confined to narrow domains or restricted task scopes. This systemic limitation underscores the urgent demand for a comprehensive, multi-dimensional benchmark specifically designed to stress-test the full spectrum of time series reasoning.

In this paper, we introduce TSRBENCH, a large-scale and comprehensive benchmark curated to assess the time series problem-solving capability of generalist models across multiple domains and tasks. TSRBENCH extensively collects, selects, and synthesizes problems from 14 domains. This extensive collection has culminated in a benchmark comprising 4125 problems. We categorize the problems into 4 major

*Table 1.* Comparison with Representative Time Series Benchmarks. *Modality* denotes the input format, where **T** and **V** represent textual and visual time series, respectively.

| Benchmark | Scale & Diversity | | | | Reasoning Capabilities | | | | Modality |
|---|---|---|---|---|---|---|---|---|---|
| | # Domains | # Tasks | # Questions | Multivariate | Perc. | Reas. | Pred. | Dec. | |
| *Forecasting-Centric* | | | | | | | | | |
| TimeMMD (Liu et al., 2024) | 9 | 1 | 16K | ✓ | ✗ | ✗ | ✓ | ✗ | **T** |
| CiK (Williams et al., 2024) | 8 | 1 | 0.3K | ✗ | ✗ | ✗ | ✓ | ✗ | **T** |
| *Analysis-Centric* | | | | | | | | | |
| TimeSeriesExam (Cai et al., 2024a) | 1 | 5 | 0.7K | ✗ | ✓ | ✗ | ✗ | ✗ | **T**, **V** |
| MTBench (Chen et al., 2025b) | 2 | 4 | 2.4K | ✗ | ✗ | ✓ | ✗ | ✗ | **T** |
| EngineMT-QA (Wang et al., 2025c) | 1 | 4 | 11K | ✓ | ✓ | ✓ | ✗ | ✓ | **V** |
| SciTS (Wu et al., 2025) | 12 | 7 | 51K | ✓ | ✓ | ✗ | ✓ | ✗ | **T** |
| TimeMQA (Kong et al., 2025) | 12 | 5 | 200K | ✗ | ✓ | ✗ | ✗ | ✗ | **T** |
| TSR-SUITE (Guan et al., 2025) | 9 | 4 | 4K | ✗ | ✗ | ✓ | ✓ | ✓ | **T** |
| **TSRBench (Ours)** | **14** | **15** | **4.1K** | ✓ | ✓ | ✓ | ✓ | ✓ | **T, V, T+V** |

dimensions of time series abilities-Perception, Reasoning, Prediction, and Decision-Making, which comprise 15 tasks for different abilities (See Figure 1 for examples). Additionally, it supports four modalities of time series for generalist models: text, image, text-image interleaved, and time series embeddings, providing a comprehensive evaluation and comparison that modern AI systems could handle.

To facilitate the evaluation of Large Language Models (LLMs) and Multimodal Large Language Models (MLLMs), we design a unified evaluation setup. Time series are transformed into textual sequences of numbers for LLMs, plots for VLMs, and embeddings for Time Series LLMs. For proprietary models, we evaluate text-form (T), vision-form (V), and a combined (T+V) representation to test modality fusion. Based on TSRBENCH, we evaluate 6 leading proprietary models (e.g., GPT-5 (OpenAI, 2025)), 13 open-source LLMs (e.g., Qwen3 (Yang et al., 2025a)), 13 open-source VLMs (e.g., InternVL3.5 (Wang et al., 2025b)), and 4 open-source TSLLMs (e.g., TS-Reasoner (Yu et al., 2025b)). Our evaluation yields four key findings: i) While current generalist models demonstrate strong performance on time series perception, they struggle significantly with complex reasoning, forecasting, and decision-making tasks. ii) The scaling law holds for most time series reasoning tasks on both LLMs and VLMs, with the notable exception of time series prediction. iii) Time Series Prediction tasks have weak relationships with other tasks. iv) Textual and visual representations of time series are strongly complementary, often solving different sets of problems, yet current models struggle to leverage both modalities simultaneously for improved performance. Additionally, our ablation studies provide practical insights into model design, particularly regarding the impact of visualization resolution, tool augmentation, and inference-time reasoning effort. These findings and analyses highlight key bottlenecks of current generalist models and provide valuable insights for the design of future time series reasoning models and methods.

## 2. Related Work

**Time Series Benchmarks.** Time series has long been studied. In the long run, existing benchmarks primarily focus on time series analysis tasks, including forecasting (Godahewa et al., 2021; Bauer et al., 2021; Qiu et al., 2024; Wang et al., 2024b; Li et al., 2025; Hu et al., 2025), classification (Ismail Fawaz et al., 2019; Ruiz et al., 2020), imputation (Du et al., 2024b; Kazijevs & Samad, 2023), and anomaly detection (Lai et al., 2021; Wenig et al., 2022; Zhou & Yu, 2024). Recent works begin to explore whether LLMs/MLLMs can understand the time series (Tan et al., 2024; Merrill et al., 2024). TimeSeriesExam (Cai et al., 2024a) evaluates the time series understanding of LLMs and VLMs through synthetic data, but only focuses on holistic perception. MTBench (Chen et al., 2025b) combines news reports with time series to assess models' reasoning capabilities, but is restricted to narrow domains such as finance and weather. TimeMMD (Liu et al., 2024) and CiK (Williams et al., 2024) focus on the time series forecasting task with the aid of contextual events or background. TSR-SUITE (Guan et al., 2025) and EngineMT-QA (Wang et al., 2025c) only cover narrow reasoning tasks, and TimeMQA (Kong et al., 2025) evaluates LLMs mainly on traditional time series analysis.

**General Reasoning Benchmarks.** Numerous benchmarks have been developed to evaluate the general reasoning and problem-solving capabilities of generalist models. Notable examples include MMMU (Yue et al., 2024) and MMMU-Pro (Yue et al., 2025), GPQA (Rein et al., 2024), which assess knowledge across a wide range of subjects. Benchmarks in science domains (Zhao et al., 2025; Wang et al., 2025a; He et al., 2024; Xu et al., 2025b; Zou et al., 2024), and engineering domains (Syed et al., 2024; Kevian et al., 2024; Guo et al., 2025b) evaluate problem-solving ability. In addition, benchmarking in social scenarios (Le et al., 2019; Kim et al., 2023; Yu et al., 2025a) evaluates the ability to understand human minds. In multimodal domains, bench-

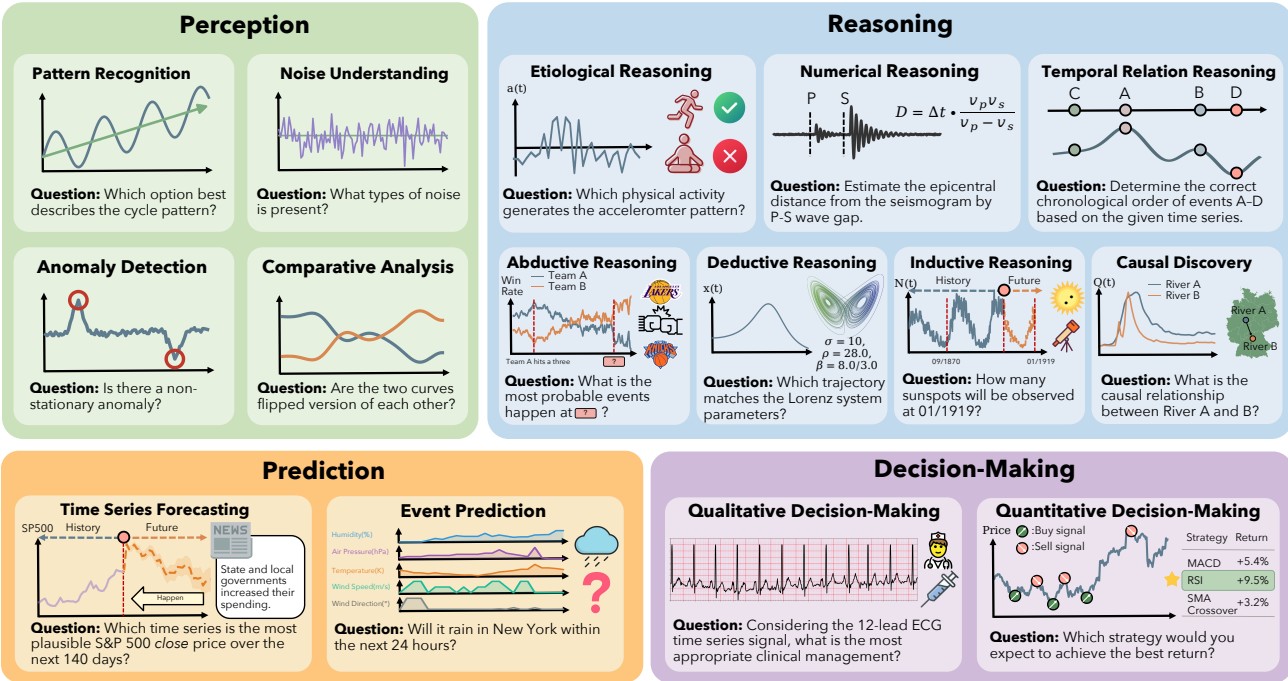

*Figure 1.* Overview of TSRBᴇɴᴄʜ. TSRBᴇɴᴄʜ evaluates generalist models across four core capabilities: Perception, Reasoning, Prediction, and Decision-Making.

marks range from scientific domains (Lu et al., 2023; Wang et al., 2024a) to embodied reasoning (Yang et al., 2025b; Du et al., 2024a) and video reasoning (Li et al., 2024; Cheng et al., 2025). While general reasoning benchmarks may sporadically incorporate time series-related tasks, they lack a comprehensive and systematic evaluation framework dedicated to temporal dynamics. We introduce TSRBᴇɴᴄʜ to fill this critical gap.

## 3. TSRBᴇɴᴄʜ

### 3.1. Overview of TSRBᴇɴᴄʜ

We introduce TSRBᴇɴᴄʜ, a comprehensive benchmark comprising 4125 instances and 15250 time series channels across 14 domains to assess generalist models on 4 key dimensions in time series reasoning: **Perception**, **Reasoning**, **Prediction**, and **Decision-Making**. Figure 2 provides a visual overview of our taxonomy and task distribution. See domain distribution in Figure 11 and § G.1 for more cases.

### 3.2. Time Series Perception

This dimension evaluates models on recognizing fundamental temporal patterns and properties of time series, covering four tasks: i) *Pattern Recognition* ii) *Noise Understanding* iii) *Anomaly Detection*, and iv) *Comparative Analysis*.

***Pattern Recognition (PR)*** evaluates the model's ability to discern fundamental time series properties, encompassing structural characteristics such as trend, cyclicity, stationarity,

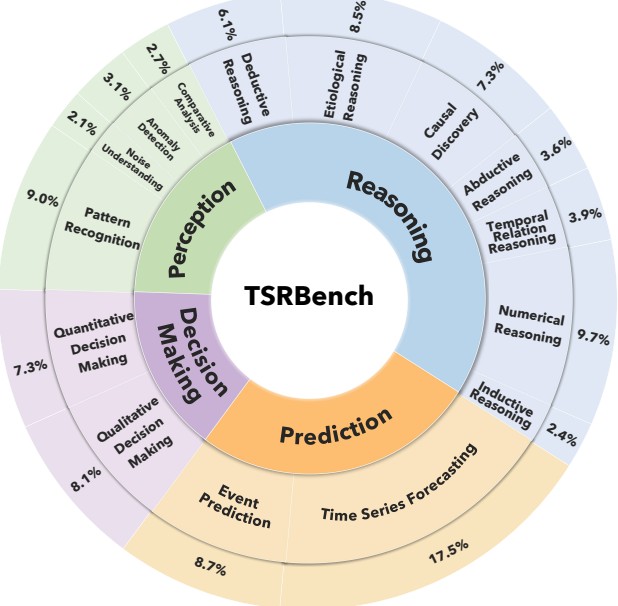

*Figure 2.* Statistics of tasks in TSRBᴇɴᴄʜ.

and core statistical attributes like the series mean. ***Noise Understanding (NU)*** challenges the model to quantify and characterize the scale and magnitude of stochastic noise inherent in the data. ***Anomaly Detection (AD)*** probes the model's capacity to identify and classify out-of-distribution observations; this extends beyond mere localization (e.g., start, middle, end) to include the characterization of anomaly

types (e.g., pattern cutoffs, signal flips) and the conceptual grasp of the underlying pattern without the perturbation. Finally, *Comparative Analysis (CA)* assesses the model's proficiency in comparative reasoning between two or more series, determining shared patterns, congruent statistical properties (e.g., variance or noise profiles), commonality in underlying data distributions, or coherence in trend direction.

### 3.3. Time Series Reasoning

This dimension evaluates the ability to derive conclusions from temporal patterns and prior knowledge, covering seven tasks: i) *Etiological Reasoning*, ii) *Causal Discovery*, iii) *Abductive Reasoning*, iv) *Temporal Relation Reasoning*, v) *Numerical Reasoning*, vi) *Deductive Reasoning*, vii) *Inductive Reasoning*.

*Etiological Reasoning (ER)* involves inferring the generative sources or underlying causal factors responsible for an observed time series. Questions in this task require both time series perception and commonsense reasoning ability. *Causal Discovery (CD)* focuses on determining the existence and direction of causal relationships between multiple time series. The causal relationship is determined by both the patterns of time series and contextual backgrounds. *Abductive Reasoning (AR)* evaluates a model's ability to infer the most plausible latent event that explains a change in the time series, given both historical and future observations. This task requires the model not only to localize the timestamp at which the event occurs, but also to detect the resulting change in the series and generate a reasonable explanatory hypothesis for it. *Temporal Relation Reasoning (TR)* challenges the model to determine the order of events based on the observations on time series. This task evaluates the localization of events on time series and establishes the correct chronological sequence of events embedded within a time series. *Numerical Reasoning (NR)* challenges the model's ability to perform quantitative calculations that require a contextual understanding of the time series domains. *Deductive Reasoning (DR)* requires the model to derive logically consistent conclusions from predefined rules, which requires the model to accurately apply the rules to the time series to draw the final conclusion. *Inductive Reasoning (IR)* evaluates the model's ability to infer principles or rules (e.g., periodicity) and patterns based on historical observations and domain knowledge. After that, the model needs to apply the inferred rules to predict future events at a specific time. Unlike standard forecasting, which prioritizes minimizing numerical error through curve-fitting, IR requires models to abstract the underlying rules to predict specific future events. These seven tasks comprehensively assess the model's capacity to interpret complex temporal dynamics, infer underlying causal structures, and apply logical principles to time series data beyond pattern recognition.

### 3.4. Time Series Prediction

We evaluate predictive capabilities through two tasks: i) *Time Series Forecasting* and ii) *Event Prediction*.

*Time Series Forecasting (TSF)* evaluates the prediction of future numerical values conditioned on both historical observations and contextual events. This challenges the model to reason about the interaction dynamics between the continuous series and the discrete events. To reduce the difficulty of directly predicting numerical series for generalist models (Tan et al., 2024), we formulate the forecasting tasks as multiple-choice questions. *Event Prediction (EP)* involves anticipating future discrete events given the historical time series. This requires synthesizing pattern analysis with commonsense or domain-specific reasoning to predict what events will happen in the future.

### 3.5. Time Series Decision-Making

This dimension assesses the ability of models to make decisions based on the understanding of both time series and context. We assess this through two aspects: i) *Qualitative Decision-Making* and ii) *Quantitative Decision-Making*.

*Qualitative Decision-Making (QualDM)* requires the model to leverage pattern analysis within time series and contextual knowledge to inform decisions. This task assesses the model's ability to make correct decisions under complex time series and knowledge. *Quantitative Decision-Making (QuantDM)* challenges the model to determine an optimal course of action by evaluating the outcomes of multiple possible operational procedures. This task assesses the model's ability to accurately simulate the application of distinct sets of rules and environmental constraints to a given time series. It requires the model to quantitatively compare the resulting performance metrics from each procedure and identify the single procedure that yields the optimal result.

### 3.6. Data Collection Principles for TSRBENCH

To accomplish a high-quality dataset, we have the following considerations when collecting data: (1) **High Text-Timeseries Alignment**. The context should be highly aligned with the time series and complement the information, and be indispensable for reasoning. (2) **Domain Diversity and Generalizability**. The data should be sourced from a wide array of domains to ensure the benchmark tests for general reasoning capabilities and prevent models from succeeding via domain-specific overfitting. (3) **Verifiable and Unambiguous Ground Truth**. To ensure correctness, we employ two strategies for answer generation: i) using high-fidelity simulations (e.g., Python code) where the ground-truth is unambiguously determined, and ii) retrieving and extracting from the time series or its contexts. (4) **Synthetic Data for Quantitative Reasoning**.

While real-world data provides essential complexity, it often lacks the high-precision ground truth required for rigorous quantitative evaluation. To bridge this gap, we incorporate synthetic data via simulations (e.g., chaotic physical systems and algorithmic trading backtesting). This approach ensures that the underlying data-generating processes are unambiguously determined, providing a verifiable and noise-free platform to stress-test a model's capacity for precise numerical reasoning and deductive logic. More details of data construction are provided in Appendix D.

## 4. Experiments

### 4.1. Experimental Setups.

We evaluate both open-source and proprietary LLMs, VLMs, and Time Series LLMs, including 6 proprietary models, and 30 SOTA open-source models. For proprietary models, we evaluate DeepSeek-V3.2 (Guo et al., 2025a), Gemini-2.5-Flash (Comanici et al., 2025), Claude-4.5-Haiku (PBC, 2025), o4-mini, GPT-5-mini, and GPT-5. Open-source LLMs include Qwen2.5 (3B / 7B / 72B), Qwen3 (1.7B / 8B / 32B / 235B-A22B) (Yang et al., 2025a), Gemma3 (12B / 27B) (Team et al., 2025b), InternLM3 (8B) (Cai et al., 2024b), GPT-OSS (20B / 120B) (Agarwal et al., 2025), and TimeOmni-1-7B (Guan et al., 2025). Open-source VLMs include Llama-4-Scout-17B-16E-Instruct (Platforms, 2025), Qwen2.5-VL (3B / 7B / 72B) (Bai et al., 2025), Qwen3-VL (8B / 32B / 235B-A22B) (Yang et al., 2025a), Phi4-Multimodal-8B (Abouelenin et al., 2025), InternVL3.5 (1B / 8B / 38B) (Wang et al., 2025b), MiniCPM-V-4.5 (8B) (Yao et al., 2024), and MiMo-VL-7B-RL (Team et al., 2025a). For TSLLMs, we evaluate two OpenTSLM (Langer et al., 2025) variants trained for time series QA, as well as TS-Reasoner (7B) (Yu et al., 2025b) and ChatTS (14B) (Xie et al., 2024). We enable reasoning for all the models. Time series are transformed into textual sequences of numbers for LLMs, plots via code for VLMs, and embeddings via model projectors for TSLLMs. Specifically, we render each time series as a line chart, where single-series samples occupy one figure and multivariate samples are stacked as vertical subplots sharing a common time axis. Each subplot has grid lines always enabled and is annotated with its series name. Based on our ablation study (§ 4.4), we fix the resolution at 100 PPI to balance token efficiency with feature visibility. For proprietary models, we evaluate them using textual, visualized, and combined (T+V) inputs. We use accuracy as the primary metric in our experiments.

### 4.2. Main Results

Table 2 summarizes the results for all tasks. Overall, current generalist models demonstrate strong performance on time series perception but struggle with reasoning, prediction, and decision-making tasks. Among proprietary models, we

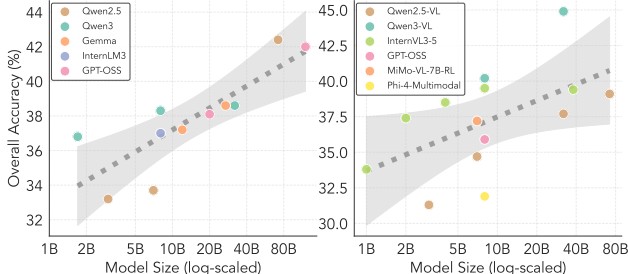

*Figure 3.* The scatter plots related to overall accuracy and model sizes. Each plot illustrates the relationship between the log-scaled number of parameters and the performance across all models. The left corresponds to LLMs and the right is VLMs.

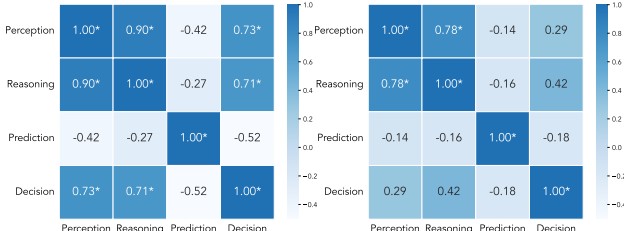

*Figure 4.* Spearman's rank correlation ($\rho$) between tasks. "(*)" marks correlations with p-values $\leqslant 0.05$.

observe that GPT-5 (T+V), which uses both textual and visual time series, achieves the highest overall accuracy (55.6%). For open-source LLMs, Qwen2.5-72B delivers the strongest overall performance (42.4%). For open-source VLMs, Qwen3-VL-32B achieves the highest overall accuracy (44.9%). Time series LLMs perform competitively with similar-sized LLMs and VLMs, yet still have a large gap from advanced models. Additionally, reasoning efforts significantly improve performance, as evidenced by the overall accuracy increases for o4-mini-high (+4.3%) and GPT-5-mini-high (+7.2%) compared to their respective baseline (T+V) models. Nevertheless, a substantial performance gap remains between the top proprietary model (GPT-5 (T+V) at 55.6%) and the best-performing open-source models (Qwen3-VL-32B at 44.9%).

### 4.3. Further Findings

> **Finding 1.** The scaling law still holds for most of the time series reasoning tasks on both LLMs and VLMs, except for time series prediction.

We quantitatively analyze the relationship between model scale and performance on time series reasoning tasks by calculating Spearman's rank correlation for both i) LLMs and ii) VLMs. Figure 3 illustrates the performance trends across model families, revealing a clear positive correlation between overall accuracy and model size. To provide deeper insight, we further investigate this correlation across indi-

*Table 2.* Model performance on TSRBENCH. Backgrounds indicate proprietary (blue) or open-source (green) models. For proprietary models, "T" denotes inputting textual time series, "V" denotes visualized time series, and "T+V" denotes using both. o4-mini and GPT-5(-mini) default to low-reasoning; the "-high" suffix denotes high-reasoning. Bold indicates best results.

| Model | Perception | | | | Reasoning | | | | | | | Prediction | | Decision | | Overall |
|---|---|---|---|---|---|---|---|---|---|---|---|---|---|---|---|---|
| | PR | NU | AD | CA | ER | CD | AR | TR | NR | DR | IR | TSF | EP | QualDM | QuantDM | |
| *Textual Time Series as Input* | | | | | | | | | | | | | | | | |
| DeepSeek-V3.2 (T) | 67.7 | 56.3 | 57.4 | 64.6 | 32.0 | 35.7 | 70.7 | 19.4 | 27.3 | 47.6 | 24.0 | 26.5 | 47.2 | 33.1 | 28.3 | 39.1 |
| o4-mini (T) | 73.1 | 65.5 | 61.2 | 71.7 | 42.6 | 39.0 | 58.0 | 34.4 | 64.7 | 42.0 | 43.0 | 33.1 | 73.3 | 30.4 | 36.0 | 47.7 |
| GPT-5-mini (T) | 72.2 | 59.8 | 63.6 | 69.9 | 45.1 | 27.7 | 62.7 | 39.4 | 65.2 | 41.3 | 50.0 | 23.5 | 67.8 | **35.5** | 30.3 | 46.6 |
| GPT-5 (T) | 75.7 | 62.1 | 61.2 | 69.0 | 42.3 | 55.3 | 85.3 | **68.8** | **72.2** | 50.0 | 62.0 | 37.8 | **79.7** | 31.9 | 32.0 | 55.5 |
| Qwen2.5-3B | 46.4 | 51.7 | 33.3 | 51.3 | 20.0 | 25.0 | 38.0 | 21.2 | 34.8 | 29.2 | 19.0 | 31.4 | 58.3 | 22.7 | 24.0 | 33.2 |
| Qwen2.5-7B | 50.7 | 50.6 | 41.1 | 54.9 | 15.1 | 32.3 | 46.0 | 28.1 | 33.2 | 24.0 | 28.0 | 33.5 | 37.5 | 31.0 | 25.7 | 33.7 |
| Qwen2.5-72B | 55.3 | 55.2 | 47.3 | 66.4 | 20.9 | 58.0 | 62.0 | 33.8 | 38.2 | 36.8 | 40.0 | 30.7 | 70.3 | 34.0 | 30.3 | 42.4 |
| Qwen3-1.7B | 45.8 | 59.8 | 38.0 | 51.3 | 18.3 | 30.1 | 48.7 | 30.0 | 27.0 | 28.4 | 20.0 | 39.2 | 67.8 | 31.3 | 24.7 | 36.8 |
| Qwen3-8B | 51.8 | 56.3 | 48.8 | 56.6 | 14.0 | 29.3 | 57.3 | 23.1 | 36.5 | 23.6 | 33.0 | 46.7 | 48.9 | 34.9 | 28.7 | 38.3 |
| Qwen3-32B | 54.7 | 52.9 | 50.4 | 62.8 | 22.6 | 35.9 | 66.0 | 27.5 | 37.5 | 34.4 | 36.0 | 31.1 | 46.1 | 34.9 | 31.7 | 38.6 |
| Qwen3-235B-A22B | 66.0 | 56.3 | 59.7 | 67.3 | 22.6 | 34.7 | 86.7 | 28.1 | 44.8 | 49.2 | 39.0 | 29.0 | 48.9 | 34.8 | 30.0 | 42.2 |
| Gemma3-12B-it | 51.5 | 59.8 | 48.9 | 65.5 | 21.1 | 34.7 | 42.0 | 23.7 | 33.2 | 28.0 | 41.0 | 39.4 | 34.2 | 34.3 | 31.3 | 37.2 |
| Gemma3-27B-it | 56.1 | 59.8 | 48.1 | 68.1 | 22.3 | 37.0 | 66.7 | 21.9 | 36.0 | 29.3 | 35.0 | 36.1 | 42.8 | 33.1 | 30.3 | 38.6 |
| InternLM3-8B | 51.2 | 42.5 | 35.7 | 54.0 | 23.4 | 24.7 | 48.0 | 23.8 | 30.0 | 28.7 | 31.0 | 36.6 | 66.4 | 33.7 | 26.7 | 37.0 |
| GPT-OSS-20B | 64.2 | 58.6 | 50.4 | 69.0 | 31.4 | 32.0 | 48.7 | 17.5 | 43.3 | 20.7 | 43.0 | 27.4 | 37.2 | 34.9 | 32.7 | 38.1 |
| GPT-OSS-120B | 66.8 | 56.3 | 59.8 | 69.0 | 39.1 | 26.3 | 58.7 | 31.3 | 48.5 | 28.0 | 40.0 | 31.1 | 59.7 | 33.7 | 31.0 | 42.0 |
| TimeOmni-1-7B | 55.0 | 59.8 | 41.9 | 64.6 | 28.0 | 35.3 | 46.7 | 24.4 | 31.5 | 22.8 | 34.0 | 30.0 | 49.4 | 34.0 | 30.3 | 36.7 |
| *Visual Time Series as Input* | | | | | | | | | | | | | | | | |
| o4-mini (V) | 77.4 | 72.4 | 66.7 | 73.5 | 29.7 | 30.7 | 55.3 | 39.4 | 56.2 | 32.8 | 51.0 | 29.4 | 73.6 | 29.9 | 42.3 | 46.6 |
| GPT-5-mini (V) | 78.7 | 69.0 | 69.0 | 69.0 | 36.0 | 41.0 | 58.7 | 41.9 | 54.5 | 39.6 | 51.0 | 27.5 | 66.1 | 26.0 | 28.3 | 46.0 |
| GPT-5 (V) | 83.6 | 72.4 | **69.8** | 73.5 | 34.6 | 37.0 | 81.3 | 68.1 | 64.0 | 48.0 | 57.0 | 38.8 | 71.9 | 26.9 | 30.0 | 52.4 |
| Qwen2.5-VL-3B | 44.2 | 50.6 | 41.9 | 53.1 | 23.4 | 24.0 | 42.0 | 25.6 | 30.2 | 27.6 | 25.0 | 24.2 | 44.2 | 31.0 | 20.0 | 31.3 |
| Qwen2.5-VL-7B | 48.2 | 55.2 | 42.6 | 60.2 | 21.7 | 25.3 | 57.3 | 26.9 | 30.8 | 26.0 | 32.0 | 30.7 | 46.7 | 33.7 | 26.3 | 34.7 |
| Qwen2.5-VL-72B | 60.9 | 55.2 | 55.8 | 77.0 | 30.6 | 52.0 | 58.0 | 30.6 | 34.0 | 35.6 | 32.0 | 22.6 | 46.1 | 29.3 | 31.7 | 39.1 |
| Qwen3-VL-8B | 60.4 | 58.6 | 51.9 | 61.1 | 27.7 | 38.8 | 50.0 | 24.4 | 39.9 | 27.6 | 38.0 | **50.7** | 23.9 | 33.4 | 29.7 | 40.2 |
| Qwen3-VL-32B | 73.9 | 59.8 | 59.7 | 76.1 | 31.7 | 44.7 | 69.3 | 39.4 | 56.6 | 41.7 | 45.0 | 37.4 | 24.4 | 33.7 | 35.7 | 44.9 |
| Qwen3-VL-235B-A22B | 65.8 | 65.5 | 61.2 | 71.7 | 39.7 | 47.7 | 84.7 | 25.0 | 43.0 | 42.8 | 31.0 | 25.3 | 26.4 | 32.5 | 28.7 | 41.0 |
| Phi4-Multimodal-8B | 52.3 | 46.0 | 41.9 | 48.7 | 24.6 | 22.3 | 28.7 | 23.1 | 30.5 | 25.2 | 26.0 | 32.1 | 25.8 | 34.9 | 29.0 | 31.9 |
| Llama-4-Scout-17B-16E | 41.5 | 44.8 | 38.8 | 46.9 | 25.1 | 53.0 | 75.3 | 28.1 | 41.5 | 30.0 | 23.0 | 39.7 | 78.0 | 33.7 | 33.3 | 42.3 |
| InternVL3.5-1B | 47.2 | 49.4 | 38.8 | 46.9 | 25.7 | 24.3 | 46.7 | 24.4 | 29.2 | 24.0 | 27.0 | 34.7 | 46.4 | 34.3 | 21.3 | 33.8 |
| InternVL3.5-8B | 60.9 | 55.2 | 52.7 | 64.6 | 25.7 | 38.3 | 60.0 | 27.5 | 40.5 | 31.6 | 29.0 | 38.2 | 41.9 | 33.4 | 22.3 | 39.5 |
| InternVL3.5-38B | 58.8 | 60.9 | 55.8 | 69.9 | 30.9 | 43.0 | 52.7 | 31.2 | 43.8 | 30.4 | 34.0 | 32.2 | 32.5 | 35.2 | 29.0 | 39.4 |
| MiniCPM-V-4.5-8B | 63.6 | 56.3 | 56.6 | 62.8 | 22.3 | 24.3 | 54.7 | 20.6 | 26.5 | 22.8 | 24.0 | 44.6 | 27.8 | 26.3 | 29.7 | 35.9 |
| MiMo-VL-7B-RL | 58.2 | 64.4 | 52.7 | 65.5 | 23.7 | 34.7 | 65.3 | 30.6 | 33.0 | 25.6 | 36.0 | 35.0 | 29.7 | 29.0 | 33.0 | 37.2 |
| *Both Visual & Textual Time Series as Input* | | | | | | | | | | | | | | | | |
| Claude-4.5-Haiku (T+V) | 57.1 | 41.4 | 48.8 | 55.8 | 24.0 | 19.7 | 78.0 | 32.5 | 50.0 | 45.2 | 52.0 | 22.2 | 48.3 | 18.1 | 28.0 | 37.1 |
| Gemini-2.5-Flash (T+V) | 75.2 | 73.6 | 69.0 | **79.6** | 36.0 | 31.0 | 86.7 | 35.6 | 39.8 | 36.8 | 43.0 | 26.7 | 66.7 | 34.3 | **49.7** | 46.5 |
| o4-mini (T+V) | 79.0 | 70.1 | 65.1 | 76.1 | 37.4 | 38.0 | 64.0 | 35.0 | 66.8 | 24.8 | 51.0 | 29.3 | 74.4 | 29.9 | 36.0 | 48.2 |
| o4-mini-high (T+V) | 82.5 | 71.3 | 64.3 | 76.1 | 44.0 | 66.3 | 76.7 | 55.0 | 70.8 | 30.8 | 53.0 | 33.6 | 71.9 | 25.4 | 24.3 | 52.5 |
| GPT-5-mini (T+V) | 78.4 | 65.5 | 63.6 | 72.6 | 38.9 | 36.3 | 76.0 | 33.1 | 66.2 | 30.0 | 54.0 | 24.4 | 65.6 | 34.3 | 29.7 | 46.9 |
| GPT-5-mini-high (T+V) | 80.3 | **75.9** | 63.6 | 75.2 | **47.4** | **69.0** | 82.0 | 56.2 | 71.8 | 42.0 | 57.0 | 32.5 | 68.9 | 32.8 | 24.3 | 54.1 |
| GPT-5 (T+V) | **84.9** | 72.4 | 61.2 | 70.8 | 38.9 | 50.3 | **93.3** | **68.8** | 72.0 | 44.0 | 58.0 | 37.8 | 78.1 | 31.9 | 34.3 | **55.6** |
| *Embedded Time Series as Input* | | | | | | | | | | | | | | | | |
| OpenTSLM-tsqa-sp-3B | 39.9 | 40.2 | 35.7 | 43.4 | 20.3 | 32.3 | 33.3 | 28.1 | 24.5 | 26.0 | 32.0 | 32.5 | 35.6 | 28.4 | 28.7 | 31.0 |
| OpenTSLM-tsqa-flamingo-3B | 41.2 | 41.4 | 34.1 | 38.9 | 22.9 | 30.3 | 36.0 | 21.2 | 22.5 | 28.8 | 29.0 | 28.1 | 45.6 | 25.4 | 29.7 | 30.7 |
| ChatTS-14B | 50.7 | 50.6 | 46.5 | 55.8 | 21.7 | 34.3 | 51.3 | 24.4 | 30.5 | 23.6 | 25.0 | 19.2 | 52.2 | 29.3 | 27.7 | 33.5 |
| TS-Reasoner-7B | 53.1 | 56.3 | 48.1 | 50.4 | 28.9 | 24.3 | 57.3 | 26.9 | 37.2 | 23.6 | 24.0 | 31.7 | 55.0 | 32.8 | 21.7 | 36.4 |

*Table 3.* Spearman's rank correlation ($\rho$) between LLM and VLM performances and model size on main dimensions. "(*)" marks correlations with p-values $\leqslant 0.05$.

| $\rho$ | Overall | Perception | Reasoning | Prediction | Decision-Making |
|---|---|---|---|---|---|
| LLM | 0.9248 (*) | 0.8929 (*) | 0.9795 (*) | -0.2415 | 0.7380 (*) |
| VLM | 0.6436 (*) | 0.8301 (*) | 0.6389 (*) | -0.2612 | 0.5596 |

vidual capability dimensions. As detailed in Table 3, while Perception, Reasoning, and Decision-Making exhibit strong positive correlations with model size, Prediction tasks notably diverge from this trend for both LLMs and VLMs. This discrepancy indicates that current generalist models, even when scaled and provided with context, continue to struggle with effective forecasting (Tan et al., 2024). See

Appendix F.2 on correlations for each task.

> **Finding 2.** Perception, Reasoning, and Decision are highly correlated, and have weak correlation with the Prediction task.

To investigate the correlation between tasks, we compute Spearman's rank correlation between four main dimensions, and the results are shown in Figure 4. We found that both LLMs and VLMs show a strong correlation within perception, reasoning, and decision-making tasks, but a weak correlation with prediction tasks. This indicates that even though the model can effectively understand and reason on the time series, it still falls short in forecasting the numerical time series and events.

*Table 4.* Performance changes of models with tool augmentation. Δ indicates the performance difference after enabling tool-augmented reasoning.

| Model | Perception | | | | Reasoning | | | | | | | Prediction | | Decision | | Overall |
|---|---|---|---|---|---|---|---|---|---|---|---|---|---|---|---|---|
| | PR | NU | AD | CA | ER | CD | AR | TR | NR | DR | IR | TSF | EP | QualDM | QuantDM | |
| *Proprietary models* | | | | | | | | | | | | | | | | |
| o4-mini (T+V) Δ | -1.6 | 0.0 | +1.6 | -2.6 | -1.7 | +1.3 | +5.3 | +4.4 | -1.8 | +0.8 | -3.0 | +4.6 | -3.3 | +4.7 | +5.3 | +1.2 |
| GPT-5-mini (T+V) Δ | -0.5 | +1.2 | +2.3 | -0.9 | +0.2 | -2.6 | -4.0 | +1.3 | +1.3 | -1.6 | -3.0 | +2.1 | -1.2 | +1.5 | +5.0 | +0.5 |
| GPT-5 (T+V) Δ | -2.2 | -3.4 | +3.1 | -0.9 | +1.1 | -3.3 | -2.6 | +1.2 | +2.5 | +14.0 | +4.0 | +1.4 | -0.9 | -3.2 | -1.3 | +0.6 |

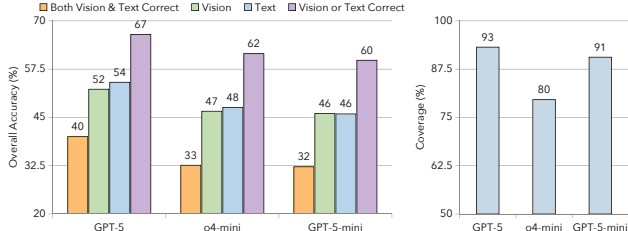

*Figure 5.* **Analysis of modality complementarity. Left:** Comparison between textual and visual time series representations. **Right:** Ratio of model (T+V) answers identical to model (T) or model (V).

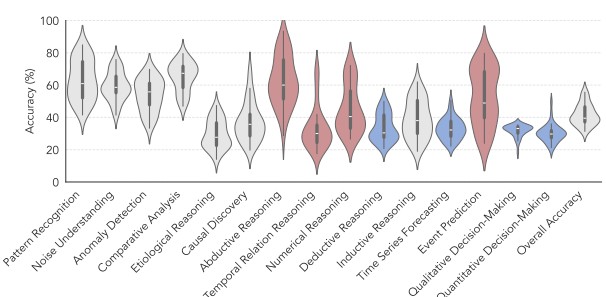

*Figure 6.* Performance distribution of evaluated models across TSRBENCH tasks. High and low inter-model variance tasks are highlighted in red and blue, respectively.

---

> **Finding 3.** Although the textual and visual representations of time series achieve similar performance, they are strongly complementary.

> **Finding 4.** Tasks with high variance, highlighted in red in Figure 6, indicate that poorer-performing models can be improved through distillation from stronger ones, while low-accuracy, low-variance tasks shown in blue reveal shared weaknesses requiring better training data.

We investigate the impact of different time series modalities, textual versus visual, using o4-mini, GPT-5-mini, and GPT-5. Our results reveal two key findings regarding modality performance and complementarity.

First, although textual and visual modalities achieve comparable overall accuracy, their strengths diverge across tasks. As shown in Table 2, visual representations outperform textual ones in Perception tasks, but this advantage diminishes in Reasoning, Prediction, and Decision-Making tasks that require fine-grained information extraction. This motivates an analysis of whether the two modalities capture complementary features. Specifically, we investigate the proportion of instances correctly solved by both representations (intersection) versus those solved by at least one (union). As shown in Figure 5 (left), the intersection yields low accuracy while the union yields high accuracy. This indicates that textual and visual representations are successful on different subsets of samples, with neither approach being dominant. However, enabling models to jointly process textual and visual time series (T+V) does not yield significant gains over single-modality inputs (Table 2). As shown in Figure 5 (right), T+V solutions largely overlap with those already solved by either modality alone, suggesting that current models fail to effectively fuse cross-modal information.

A closer examination of the performance distribution in Figure 6 reveals two distinct regimes of task difficulty. To characterize these regimes, we compute the mean accuracy and variance for each task by aggregating results across all models and input time-series representations (e.g., text and vision) reported in Table 2. On the one hand, tasks with high variance (red in Figure 6), such as Abductive Reasoning and Event Prediction, show that specific models handle them well while others struggle. This gap suggests that weaker models could likely improve their reasoning skills through knowledge distillation from the stronger ones. On the other hand, low-accuracy, low-variance tasks (blue in Figure 6), including Quantitative Decision-Making and Time Series Forecasting, show uniformly poor performance across all models. This reveals a shared weakness in current generalist models, where progress will likely require data-centric pre-training with richer quantitative and temporal supervision.

### 4.4. Further Investigation

**Does Time Series Analysis Tool-use Help?** To investigate whether generalist models fail due to their weaknesses in understanding the time series, we enrich the understanding

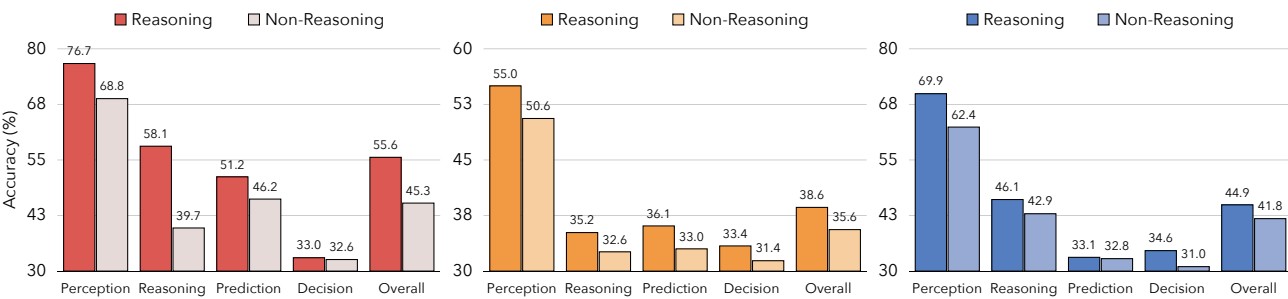

*Figure 7.* Performance of GPT-5 (T+V) (**left**), Qwen3-32B (**middle**), Qwen3-VL-32B (**right**), with (non-) reasoning modes.

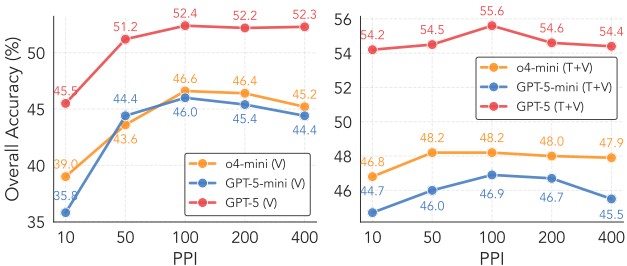

*Figure 8.* Impact of visual resolution on performance for Visual-only (**left**) and Text+Visual (**right**) as inputs.

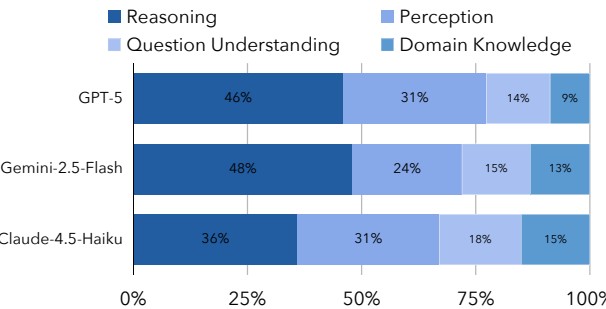

*Figure 9.* Error type distribution of three models on TSRBench.

of temporal patterns by extracting a comprehensive set of statistical, structural, and frequency-domain features from each time series. These include summary statistics (e.g., mean, variance), trend and seasonality measures, temporal dependencies, local extrema and change points, similarity metrics, and outlier indicators. This enriched representation is appended to each prompt to provide the model with explicit temporal context. We conduct experiments on o4-mini, GPT-5-mini, and GPT-5. The results in Table 4 show that the additional features yield a modest overall improvement, though gains vary considerably across tasks. This suggests that explicit time series statistics can partially compensate for models' weaknesses in temporal understanding, but are not sufficient to address them in all settings. Further details and the specific analysis functions used are provided in Appendix E.

**Impact of Resolutions for visual time series.** We investigate the effect of five visualization resolutions on performance. Our results, shown in Figure 8, indicate that mid-range resolutions (100 PPI) achieve better results compared to both lower (10 PPI, 50 PPI) and higher (200 PPI, 400 PPI) resolutions. While low-resolution images may lack the fine-grained details necessary for understanding and reasoning, excessively high resolutions can introduce unnecessary complexity, making it harder for models to focus on relevant information or capture global patterns for reasoning. Notably, the performance degradation at low resolutions is mitigated when a textual time series is provided alongside visualizations. This suggests that textual time series can partially compensate for visual information

loss, providing a degree of cross-modal redundancy. These results highlight the importance of selecting an appropriate resolution for visualized time series reasoning.

**Impact of Inference-Time Scaling.** We investigate the impact of inference-time computation on performance by evaluating GPT-5, Qwen3-32B, and Qwen3-VL-32B in both reasoning and non-reasoning modes. As illustrated in Figure 7, a clear performance divergence emerges: while Perception tasks remain robust to reduced compute, Reasoning, Prediction, and Decision-Making tasks suffer a sharp degradation. These results suggest that while generalist models can intuitively "perceive" temporal patterns through fast, heuristic processing, deriving logical conclusions from those patterns is a computationally intensive process that necessitates deliberative reasoning.

### 4.5. Error Analysis

To systematically diagnose failure mechanisms, we conducted a fine-grained error analysis on GPT-5 (T+V), Gemini-2.5-Flash (T+V), and Claude-4.5-Haiku (T+V). We randomly sampled 150 failure instances (10 per task subset) and categorized them into a four-tier taxonomy: Reasoning, Perception, Question Understanding, and Domain Knowledge. Our quantitative analysis reveals that Reasoning and Perception errors are the predominant failure modes across all models, whereas errors stemming from Question Understanding or Domain Knowledge misapplication are marginal. This distribution pinpoints the critical bottleneck: current models are limited not by a lack of knowledge or linguistic

comprehension, but by deficiencies in perceiving temporal patterns and performing rigorous reasoning based on those perceptions. See Appendix G.2 for error cases.

## 5. Conclusion

We introduce TSRBENCH, a comprehensive benchmark for systematically evaluating the time-series understanding and reasoning capabilities of generalist models across diverse tasks and domains. Our extensive empirical study reveals several fundamental challenges, which highlight critical limitations of current generalist models. We hope TSRBENCH will inspire and guide future research toward building more capable time series reasoning approaches and models.

## Impact Statement

TSRBENCH aims to provide a comprehensive benchmark for complex reasoning on time series in future research. Time series plays a critical role in real-world applications; therefore, enhancing the temporal reasoning capabilities of generalist models has profound implications for high-stakes domains such as healthcare, finance, and industrial systems. By providing a standardized platform to evaluate perception, reasoning, prediction, and decision-making, our work contributes to identifying the blind spots in current AI systems and provides potential research directions on time series reasoning.

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

# A. Additional Related Work

### A.1. MLLM/LLM Reasoning.

Recent advances in Large Language Models (LLMs) and Multimodal LLMs (MLLMs) have catalyzed a shift from pattern matching to deliberate problem-solving. Initial efforts focused on prompt engineering to induce intermediate reasoning steps (Wei et al., 2022; Khot et al., 2022; Zhou et al., 2022; Zheng et al., 2023; Zhang et al., 2023b; Gao et al., 2024; Ho et al., 2025; Chen et al., 2025c; Liang et al., 2025), though these approaches remain heavily dependent on heuristic human design. To mitigate this dependency, subsequent research has explored scaling test-time compute to enhance reasoning depth. This includes parallel sampling strategies (Brown et al., 2024; Snell et al., 2024; Karan & Du, 2025; Wang et al., 2022; Aggarwal et al., 2023), sequential refinement frameworks (Yao et al., 2022; Yang et al., 2023; Gou et al., 2023; Zhang et al., 2024c; 2023a), and hybrid search architectures (Yao et al., 2023; Besta et al., 2024; Hao et al., 2023; Ding et al., 2024). Most recently, a paradigm shift towards internalized reasoning has emerged, where models explicitly learn reasoning behaviors via Reinforcement Learning (Guo et al., 2025a; Yu et al., 2024; Xie et al., 2025; Zhou et al., 2025; Huang et al., 2025; Yu et al., 2026). This trajectory has culminated in advanced reasoning models such as the Qwen3-series (Yang et al., 2025a), which demonstrate remarkable proficiency in textual and multimodal tasks. However, the adaptation of these reasoning paradigms to the time series domain remains significantly underexplored, presenting a critical avenue for future investigation.

### A.2. Benchmarks for Generalist Models.

A variety of benchmarks assess generalist model reasoning and their problem-solving capabilities. In the early stages of generalist models, benchmarks such as GLUE (Wang et al., 2018), BERTScore (Zhang et al., 2019), and SuperGLUE (Wang et al., 2019) primarily focused on natural language understanding through small-scale, single-task evaluations. As generalist models rapidly scale up in size and begin to exhibit emergent generalization abilities, a new wave of benchmarks has emerged, such as MMMU (Yue et al., 2024), BIG-bench (Srivastava et al., 2023), Q-Bench (Wu et al., 2023), and Seed-Bench (Li et al., 2023). These benchmarks aim to assess a wider range of capabilities, including reasoning, factual knowledge, and visual recognition. To more comprehensively evaluate specific abilities, research works subdivide the aspects of evaluation, such as science reasoning (Cobbe et al., 2021; Xu et al., 2025a; Lu et al., 2023; Zhang et al., 2024b; Zhong et al., 2025), social reasoning (Guha et al., 2023; Zhang et al., 2024a; Kim et al., 2023), and engineering reasoning (Chen, 2021; Jimenez et al., 2023; Yu et al., 2018; Guo et al., 2025b; Chen et al., 2025a; Guo et al., 2025c). These diverse benchmarks highlight the broad coverage yet increasing specialization of reasoning evaluations, reflecting a shift from coarse-grained language understanding toward fine-grained, domain-specific, and comprehensive assessments of complex reasoning abilities in modern generalist models.

# B. Future Research Directions

While TSRBENCH represents a significant step forward in evaluating generalist models on time series understanding and reasoning, several challenges remain, offering rich opportunities for future research. Below, we outline potential research directions:

- **Multi-view Time Series Understanding:** Current models struggle to fuse textual and visual representations despite their strong complementarity. Future research should focus on developing alignment techniques that effectively fuse high-resolution visual patterns with a semantic textual context to enhance holistic understanding.

- **Large-scale Pretrained Time Series Models:** Given the collective blind spots in quantitative forecasting observed across generalist models, there is a critical need to develop foundation models pre-trained specifically on a massive-scale, diverse time series corpora. This data-centric approach is essential to bridge the gap between semantic reasoning and precise numerical extrapolation.

- **Multi-agent Time Series Systems:** Complex time series problems often require distinct capabilities ranging from pattern recognition to logical deduction and domain knowledge retrieval. A multi-agent framework could decompose these tasks, employing specialized agents to collaborate on and verify predictions, thereby overcoming the limitations of single-model reasoning.

- **Test-time Scaling Approaches:** Our ablation studies reveal that reasoning-intensive tasks suffer significantly without sufficient inference-time computation. In addition, we show on o4-mini and GPT-5-mini that increased reasoning

*Table 5.* Model list and URL.

| Model Name | URL |
|---|---|
| **Proprietary Models** | |
| DeepSeek-V3.2-Exp | https://huggingface.co/deepseek-ai/DeepSeek-V3.2-Exp |
| o4-mini | https://platform.openai.com/docs/models/o4-mini |
| GPT-5-mini | https://platform.openai.com/docs/models/gpt-5-mini |
| GPT-5 | https://platform.openai.com/docs/models/gpt-5 |
| Claude-4.5-Haiku | https://www.anthropic.com/claude/haiku |
| Gemini-2.5-Flash | https://cloud.google.com/vertex-ai/generative-ai/docs/models/gemini/2-5-flash |
| **Open Source Large Language Models** | |
| Qwen3-1.7B | https://huggingface.co/Qwen/Qwen3-1.7B |
| Qwen3-8B | https://huggingface.co/Qwen/Qwen3-8B |
| Qwen3-32B | https://huggingface.co/Qwen/Qwen3-32B |
| Qwen3-235B-A22B | https://huggingface.co/Qwen/Qwen3-235B-A22B-Instruct-2507 |
| Qwen2.5-3B | https://huggingface.co/Qwen/Qwen2.5-3B-Instruct |
| Qwen2.5-7B | https://huggingface.co/Qwen/Qwen2.5-7B-Instruct |
| Qwen2.5-72B | https://huggingface.co/Qwen/Qwen2.5-72B-Instruct |
| Gemma-3-12B-it | https://huggingface.co/google/gemma-3-12b-it |
| Gemma-3-27B-it | https://huggingface.co/google/gemma-3-27b-it |
| InternLM3-8B | https://huggingface.co/internlm/internlm3-8b-instruct |
| GPT-OSS-20B | https://huggingface.co/openai/gpt-oss-20b |
| GPT-OSS-120B | https://huggingface.co/openai/gpt-oss-120b |
| TimeOmni-1-7B | https://huggingface.co/anton-hugging/TimeOmni-1-7B |
| **Vision Language Models** | |
| Qwen2.5-VL-3B | https://huggingface.co/Qwen/Qwen2.5-VL-3B-Instruct |
| Qwen2.5-VL-7B | https://huggingface.co/Qwen/Qwen2.5-VL-7B-Instruct |
| Qwen2.5-VL-72B | https://huggingface.co/Qwen/Qwen2.5-VL-72B-Instruct |
| Qwen3-VL-8B | https://huggingface.co/Qwen/Qwen3-VL-8B-Instruct |
| Qwen3-VL-32B | https://huggingface.co/Qwen/Qwen3-VL-32B-Instruct |
| Qwen3-VL-235B-A22B | https://huggingface.co/Qwen/Qwen3-VL-235B-A22B-Instruct |
| Phi4-Multimodal-8B | https://huggingface.co/microsoft/Phi-4-multimodal-instruct |
| Llama-4-Scout-17B-16E | https://huggingface.co/meta-llama/Llama-4-Scout-17B-16E-Instruct |
| InternVL3.5-1B | https://huggingface.co/OpenGVLab/InternVL3_5-1B |
| InternVL3.5-8B | https://huggingface.co/OpenGVLab/InternVL3_5-8B |
| InternVL3.5-38B | https://huggingface.co/OpenGVLab/InternVL3_5-38B |
| MiniCPM-V-4.5-8B | https://huggingface.co/openbmb/MiniCPM-V-4_5 |
| MiMo-VL-7B-RL | https://huggingface.co/XiaomiMiMo/MiMo-VL-7B-RL |
| **Time Series Large Language Models** | |
| ChatTS-14B | https://huggingface.co/bytedance-research/ChatTS-14B |
| TS-Reasoner-7B | https://huggingface.co/ParadiseYu/TS-Reasoner-7B |
| OpenTSLM-tsqa-sp-3B | https://huggingface.co/OpenTSLM/llama-3.2-3b-tsqa-sp |
| OpenTSLM-tsqa-flamingo-3B | https://huggingface.co/OpenTSLM/llama-3.2-3b-tsqa-flamingo |

efforts substantially benefit reasoning performance. Future work should explore adaptive reasoning strategies, including advanced prompting techniques, structured reasoning, and self-verification.

## C. Model Versions

Table 5 lists the versions of the models and their official links used in our experiments. We accessed proprietary models through the API calls and open-source models via local deployment using vLLM (Kwon et al., 2023).

## D. Data Collection

We first introduce the key challenges in constructing TSRBENCH, followed by the construction pipeline. We then detail each component of the pipeline with corresponding data quality verification. Finally, we discuss considerations for fairness and data release.

**Challenges**. Creating a high-quality, multi-domain numerical-text series dataset presents significant challenges, encompassing the effective gathering, filtering, and alignment of useful textual data. First, textual sources are sparse. Unlike numerical

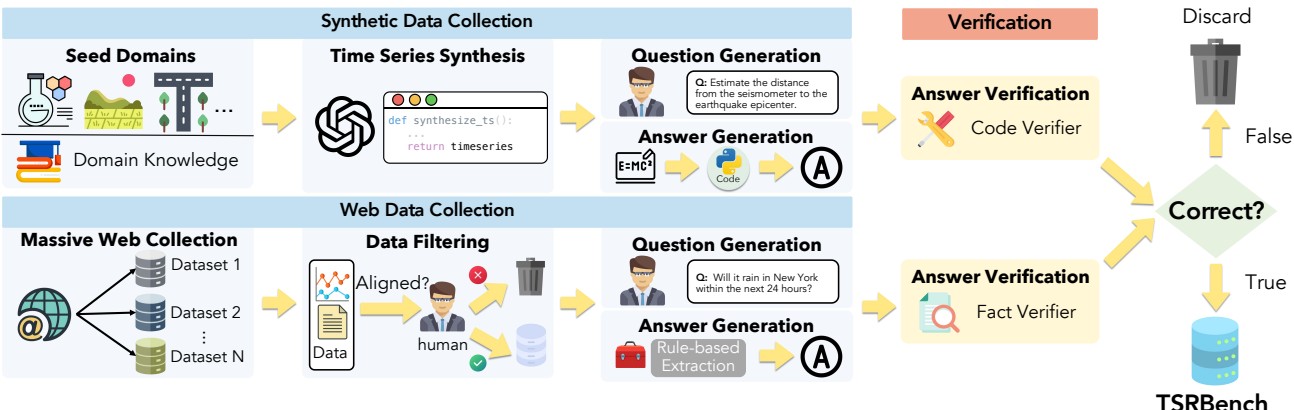

*Figure 10.* Pipeline of data collection in TSRBENCH.

data, typically provided by a packaged source, textual data are collected from a variety of dispersed sources, such as reports and news articles, necessitating extensive individual collection efforts. Second, textual information is noisy. Raw textual data often contains large portions of irrelevant information and potential data contamination, such as expert predictions in reports, requiring rigorous filtering processes to ensure data quality. Third, textual data requires precise alignment. It is essential to achieve temporal alignment between textual and numerical data by synchronizing reported times with numerical time steps (e.g., the time step where text is posted) and ensuring that the effective duration of textual information matches the relevant time frames at various granularities (e.g., a seasonal report should correspond to 12 time steps in a weekly time series). Additionally, the dataset faces challenges regarding ease of use, maintenance, and regular updates to remain relevant and useful for ongoing research and applications.

**Pipeline Overview.** We propose a comprehensive pipeline for constructing TSRBENCH. As illustrated in Figure 10, the construction process is divided into three key steps: (1) *Raw data collection.* We gather time series from reputable sources or code synthesis to ensure reliability and accuracy. (2) *Question Generation.* Questions are created by humans for each reasoning task to ensure that there is no ambiguity in the question. After that, we use either code or a rule-based extractor to generate ground-truth from the data. (3) *Verification.* For each data source, we examine whether the context is highly correlated to the time series, and further use code and rules to verify the quality of the answer to ensure the correctness.

### D.1. Data Acquisition

To address the challenges of data scarcity and noise, we employ a dual-stream strategy for raw data acquisition, ensuring both diversity and precision.

**Synthetic Data Collection.** For domains or tasks (e.g., numerical reasoning) where real-world data is sparse or difficult to isolate (e.g., complex physical simulations or specific medical scenarios), we utilize a synthesis approach. We select diverse *Seed Domains* such as chemistry and seismology. Leveraging domain knowledge, we design Python generation functions (`synthesize_ts`) to simulate realistic time series data. This approach allows us to precisely control the underlying variables, ensuring the data is clean and the parameters are known.

**Web Data Collection.** To capture real-world complexity, we aggregate massive datasets from reputable public repositories. We employ human annotators to rigorously verify the alignment between texts and time series. Crucially, we enforce strict temporal alignment to ensure that textual reports (e.g., news events) accurately correspond to numerical changes in the time series. To minimize the risk of data leakage, we process the raw data by anonymizing specific entities (e.g., replacing *Lakers* with *Team A*, *James Harden* with *Player 1*). Please refer to Table 6 for an overview of the data sources.

### D.2. Question & Answer Generation

Once the raw data is collected, we generate reasoning tasks designed to test specific analytical capabilities. To avoid the ambiguity often associated with automated generation, we manually design the question templates. We then generate answers via two methods: (1) *Code-Based Calculation:* For synthetic data, answers are derived programmatically using the underlying physical formulas or logic rules defined during generation. (2) *Rule-Based Extraction:* For web data, we use rule-based extractors to derive the correct answers directly from the data source, or retrieve them from associated metadata, time series values, or textual contexts.

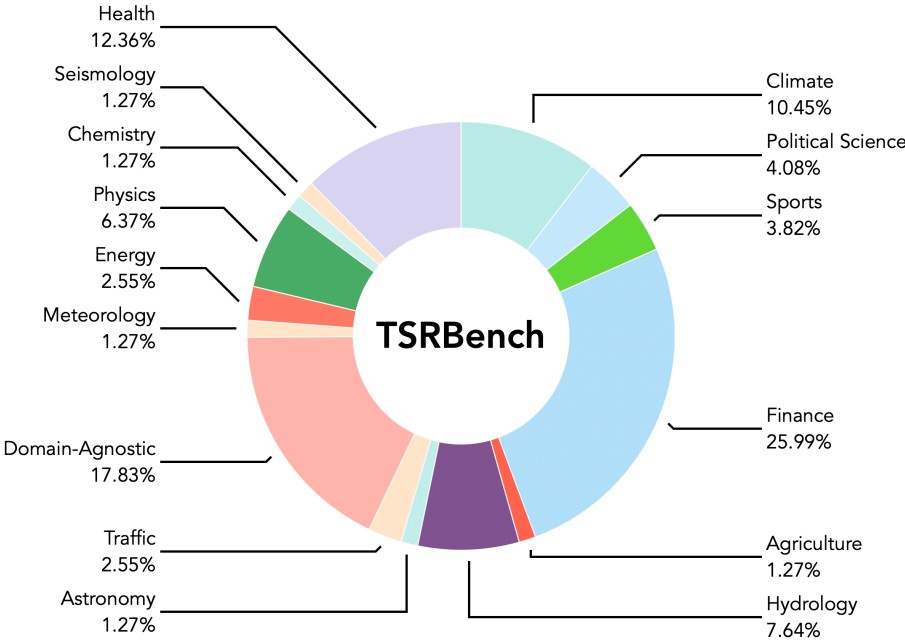

*Figure 11.* Domain distribution of TSRBENCH. *Domain-Agnostic* indicates time series that are not associated with any specific domain.

### D.3. Answer Verification

To ensure TSRBENCH serves as a rigorous benchmark, we implement a strict two-stage verification mechanism before any data is accepted. For synthetic tasks, we employ a **Code Verifier** that re-executes the generation logic to ensure the answer aligns precisely with the simulation parameters. For web-based tasks, we utilize a **Fact Verifier** to cross-reference generated answers against ground-truth records. Furthermore, we rigorously construct distractors for multiple-choice questions to ensure validity. We generate these either through algorithmic manipulation (e.g., reversing the ground-truth time series) to guarantee the choice is incorrect or by retrieving semantically distinct options validated by LLMs.

*Table 6.* Data sources for each task.

| Category | Data Source |
|---|---|
| Pattern Recognition | TimeSeriesExam (Cai et al., 2024a) |
| Noise Understanding | TimeSeriesExam (Cai et al., 2024a) |
| Anomaly Detection | TimeSeriesExam (Cai et al., 2024a) |
| Comparative Analysis | TimeSeriesExam (Cai et al., 2024a) |
| Etiological Reasoning | LEAVES (Fei et al., 2024), Human Activity Recognition (Kwapisz et al., 2011; Bachlin et al., 2009) |
| Causal Discovery | CausalRiver (Stein et al., 2025) |
| Abductive Reasoning | GAMETIME (Tan et al., 2025) |
| Temporal Relation Reasoning | Time-IMM (Chang et al., 2025) |
| Numerical Reasoning | Synthetic Data |
| Deductive Reasoning | Synthetic Data |
| Inductive Reasoning | Kaggle: Philippines Typhoon Trend (2014–2024), Sunspots Dataset |
| Time Series Forecasting | CAMFE (Zhang et al., 2025) |
| Event Prediction | TimeCAP (Lee et al., 2025) |
| Qualitative Decision-Making | ECG-QA (Oh et al., 2023), PTB-XL (Wagner et al., 2020) |
| Quantitative Decision-Making | Synthetic Data |

### D.4. Data Contamination & Quality Control

We perform data leakage detection to alleviate the potential data contamination in TSRBENCH. Following (Xu et al., 2024), we utilize n-gram accuracy to detect any data leakage within different LLMs. Concretely, we combined each problem, the textual time series, and its solution in the dataset and randomly

*Table 7.* The proportion (in %) of data leakage detection for TSRBENCH.

| Model | N-gram Accuracy (%) |
|---|---|
| o4-mini | 0.3% |
| GPT-5-mini | 0.1% |
| GPT-5 | 0.4% |

chose $K = 20$ positions for extracting 5-grams. A sample is considered contaminated if the 5-grams predicted by the model match the actual 5-grams from the dataset. We perform on the best-performing models, o4-mini, GPT-5-mini, GPT-5. The results are presented in Table 7. It is shown that most of models exhibit low N-gram accuracy, indicating a low data leakage.

## E. Time Series Analysis Tools

To enhance the reasoning capabilities of the models regarding numerical data, we integrated a deterministic analysis module. This module processes raw time series input $X = \{x_1, x_2, \ldots, x_n\}$ and injects structured statistical summaries into the model's context window alongside the raw time series. The analysis pipeline consists of five core components: statistical profiling, trend detection, extremum identification, change point detection, and series comparison.

### E.1. Data Preprocessing

Prior to analysis, all input series are sanitized. Non-numeric artifacts (e.g., formatting commas) are removed, and the series is converted to a floating-point array.

### E.2. Descriptive Statistics

We compute the fundamental statistical moments and distribution properties to provide the model with a global view of the data scale and shape. The computed metrics include the mean ($\mu$), standard deviation ($\sigma$), range ($x_{max} - x_{min}$), median, and variance ($\sigma^2$). Additionally, we calculate the skewness and kurtosis to describe the asymmetry and tailedness of the distribution, respectively, utilizing the `scipy.stats` library.

### E.3. Trend Analysis

To formally quantify the trajectory of the time series, we utilize Ordinary Least Squares (OLS) linear regression. We model the series as $x_t = \beta_1 t + \beta_0 + \epsilon$, where $t$ is the time index. The tool outputs:

- **Slope ($\beta_1$):** Indicates the direction and magnitude of the trend (increasing if $\beta_1 > 0$, decreasing if $\beta_1 < 0$).

- **Coefficient of Determination ($R^2$):** Measures the proportion of variance in the dependent variable predictable from the independent variable.

- **Trend Strength:** We categorize strength based on the Pearson correlation coefficient $r$. A trend is classified as "strong" if $|r| > 0.7$, "moderate" if $0.4 < |r| \leqslant 0.7$, and "weak" otherwise.

### E.4. Peak and Valley Detection

Local extrema are identified to highlight turning points in the data. We employ a signal processing approach (via `scipy.signal.find_peaks`) to find indices $t$ such that $x_t$ is a local maximum (peak) or minimum (valley). To filter out noise, we apply a prominence threshold $P$. By default, $P$ is dynamic:

$$P = 0.5 \cdot \sigma_X \tag{1}$$

where $\sigma_X$ is the standard deviation of the series. This ensures that only significant structural peaks are reported to the LLM, reducing context noise.

### E.5. Change Point Detection

To detect sudden shifts or volatility clustering, we analyze the first-order difference of the series, defined as $\Delta x_t = x_t - x_{t-1}$. A time step $t$ is flagged as a change point if the magnitude of the change exceeds a statistical threshold $\theta$:

$$|\Delta x_t| > \theta, \quad \text{where } \theta = 2 \cdot \text{std}(\Delta X) \tag{2}$$

This method effectively captures sudden shocks or regime changes in the time series that may be difficult for the LLM to infer from raw tokens alone.

*Table 9.* Proportion of samples containing multi-variate time series in each task.

| Perception | | | | Reasoning | | | | | | | Prediction | | Decision | | Overall |
|---|---|---|---|---|---|---|---|---|---|---|---|---|---|---|---|
| PR | NU | AD | CA | ER | CD | AR | TR | NR | DR | IR | TSF | EP | QualDM | QuantDM | Overall |
| 3.8% | 14.9% | 31.0% | 100% | 57.1% | 100% | 0% | 0% | 62.5% | 0% | 50.0% | 100% | 100% | 100% | 0% | 58.1% |

*Table 10.* Spearman's rank correlation ($\rho$) between LLM, VLM performance and model size. We mark correlations with p-values $\leqslant 0.05$ using (*).

| Metric | Perception | | | | Reasoning | | | | | | | Prediction | | Decision | |
|---|---|---|---|---|---|---|---|---|---|---|---|---|---|---|---|
| | PR | NU | AD | CA | ER | CD | AR | TR | NR | DR | IR | TSF | EP | QualDM | QuantDM |
| $\rho$ (LLM) | 0.9021 (*) | 0.1221 | 0.7352 (*) | 0.8558 (*) | 0.6469 (*) | 0.4601 | 0.6986 (*) | 0.2733 | 0.8037 (*) | 0.2831 | 0.7671 (*) | -0.5525 | -0.0046 | 0.5092 | 0.7626 (*) |
| $\rho$ (VLM) | 0.7502 (*) | 0.3812 | 0.7537 (*) | 0.7835 (*) | 0.6707 (*) | 0.7268 (*) | 0.3498 | 0.4544 | 0.6482 (*) | 0.6193 (*) | 0.3763 | 0.0280 | -0.3684 | 0.0726 | 0.5880 |

## E.6. Multivariate Comparison

When the input contains multiple time series ($X$ and $Y$), the agent performs pairwise comparisons to determine their relationship.

1. **Pearson Correlation:** We calculate the coefficient $\rho_{X,Y}$ and the associated p-value to test for linear correlation.

2. **Cross-Correlation and Lag:** We compute the normalized cross-correlation function to identify the optimal lag $\tau$ that maximizes similarity:

$$\tau_{\text{best}} = \underset{\tau}{\arg\max}(X \star Y)(\tau) \tag{3}$$

3. **Statistical Difference:** A Welch's t-test is performed to determine if the means of the two series are significantly different (p-value $< 0.05$).

## F. Additional Results

### F.1. Multivariate Coverage across Tasks

We report the proportion of multivariate[1] samples per task in Table 9. Overall, $58.1\%$ samples in TSRBENCH consist of multivariate instances, ensuring that the benchmark substantially covers reasoning over multi-channel signals in addition to univariate ones.

### F.2. Fine-grained Correlation Results

Table 10 provides a detailed correlation analysis between model size and performance on individual tasks. The results indicate that for both LLMs and VLMs, model size positively correlates with performance on most perception, reasoning, and decision-making tasks, but not on prediction tasks.

### F.3. MCQ Format Faithfully Reflects Numerical Forecasting Ability

To investigate whether the multiple-choice formulation of Time Series Forecasting reflects the true numerical forecasting ability of large models, we evaluate all LLMs and VLMs in an *open-ended* numerical forecasting setting, where models are required to generate the predicted values directly, and we use nMAE as the metric. We then compute Spearman's rank correlation between model size and forecasting performance, and contrast it with the MCQ-based correlation. The results shown in Table 8 indicate that the two formats produce highly consistent trends, suggesting that the accuracy of the MCQ faithfully reflects the quality of open-ended forecasting.

*Table 8.* Spearman's $\rho$ between model size and forecasting performance under the MCQ and open-ended settings.

| $\rho$ | MCQ | Numerical Forecasting |
|---|---|---|
| LLMs | -0.55 | -0.61 |
| VLMs | 0.03 | -0.05 |

---

[1]We use "multivariate" to denote any instance with more than one time series channel presented jointly to the model.

# G. Cases

## G.1. Question Cases

**Pattern Recognition**

**Question:** What is the most dominant pattern in this complex time series?

**Answer Choices:**
(A) Noise
(B) Trend
(C) Seasonality
**Correct Answer:** (C)

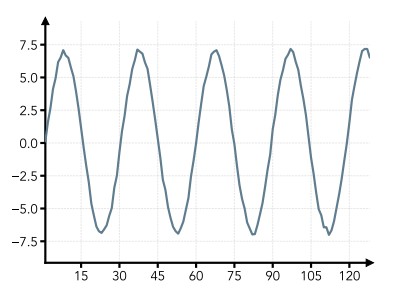

**Anomaly Detection**

**Question:** The following time series has an anomaly where the pattern is cutoff at a certain point in time. What is the likely pattern of the time series without the anomaly?

**Answer Choices:**
(A) Sine wave with linear trend
(B) Sawtooth wave with exponential trend
(C) Square wave with log trend
**Correct Answer:** (B)

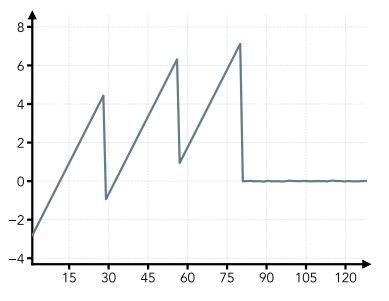

**Noise Understanding**

**Question:** The given time series is a random walk process. What is the most likely noise level?

**Answer Choices:**
(A) 0.35
(B) 8.59
(C) 3.72
**Correct Answer:** (C)

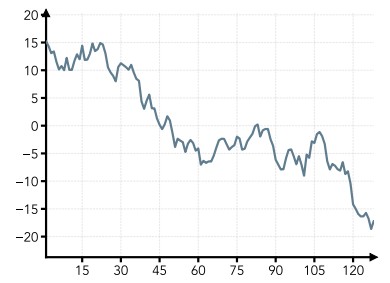

**Comparative Analysis**

**Question:** You are given two time series that both have a trend component. Do they share the same direction of trend?

**Answer Choices:**
(A) Yes
(B) No
**Correct Answer:** (A)

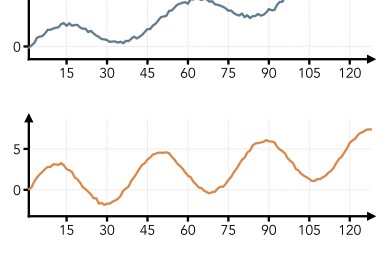

## Etiological Reasoning

**Question:** You are given a short tri-axial accelerometer time series collected at a sampling rate of 20 Hz from a subject performing a physical activity. The sequence consists of 10 consecutive timestamps (approximately 0.5 seconds), where each timestamp contains acceleration measurements along the X, Y, and Z axes, yielding a total of 30 values formatted as:

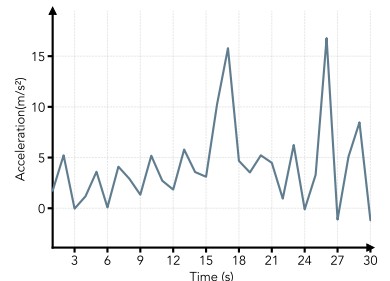

$$[x_1, y_1, z_1, x_2, y_2, z_2, \ldots, x_{10}, y_{10}, z_{10}].$$

Acceleration values lie in the range $[-20, 20]$, where 10 corresponds to $1g$ (approximately $9.8$ m/s$^2$), and the measurements include gravitational acceleration. Consequently, when the device is stationary on a flat surface, the vertical axis typically registers approximately $\pm 10$. The subject is performing one of the following six activities: *walking*, *jogging*, *sitting*, *standing*, *upstairs*, or *downstairs*. Given the provided accelerometer time series, identify the activity label that most likely generated the data.

**Answer Choices:**
(A) Walking
(B) Jogging
(C) Upstairs
(D) Downstairs
(E) Sitting
(F) Standing
**Correct Answer:** (D)

## Numerical Reasoning

**Question:** The following time series shows the concentration (in mol/L) of a reactant over time during a chemical reaction. Based on the concentration decay pattern, determine the reaction order (zero, first, or second) and calculate the rate constant $k$. For zero-order: $C = C_0 - kt$; for first-order: $C = C_0 e^{-kt}$; for second-order: $\frac{1}{C} = \frac{1}{C_0} + kt$.
Assume this is a first-order reaction and calculate $k$.

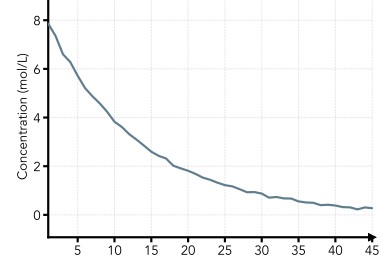

**Answer Choices:**
(A) 2.064
(B) 0.0344
(C) 0.0241
(D) 0.0758
**Correct Answer:** (B)

## Temporal Relation Reasoning

**Question:** Based on the provided 'GoldSteinScale' time series and event timestamps, determine the correct chronological order of the following events:
(1) The Australian Manufacturing Workers' Union, Australian Workers' Union, and Electrical Trades Union are involved in industrial action against Qantas, targeting the airline's engineers to demand better wages and working conditions . . .

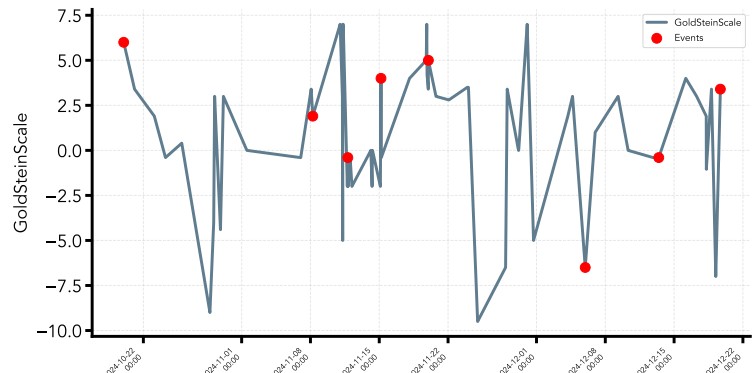

(2) The article does not directly mention labor unions or their activities. It highlights how Australian workers are increasingly using AI and robotics to reduce monotonous and physically demanding tasks, potentially leading to shifts in job roles and productivity. . . .

(3) The Australian Workers' Union (AWU) raised safety concerns prior to the incident at the Golden Plains Wind Farm, highlighting issues such as poor safety standards, non-unionized contractors, and near misses. AWU representatives expressed dissatisfaction with the company's response during a meeting, feeling their concerns were dismissed and that safety standards in the renewable sector lag behind civil construction . . .

(4) The Australian Manufacturing Workers' Union (AMWU) commissioned a report highlighting that reducing Australia's industrial gas use would protect jobs and support manufacturing sectors, emphasizing the importance of supporting workers through a transition to electrification and green hydrogen. AMWU's national secretary, Steve Murphy, stressed the need to prioritize gas resources for industries that employ hundreds of thousands of workers, as part of a fair industrial transition amid decarbonization efforts . . .

(5) Greater Victoria Canada Post workers, represented by the Canadian Union of Postal Workers (CUPW), are preparing for a potential strike after over a year of negotiations with Canada Post. The union, which includes 600 workers in the region, has voted overwhelmingly in favor of strike action, demanding a 22% wage increase over four years and other improvements such as pensionable hours and a guaranteed 40-hour work week . . .

(6) The article highlights concerns from the Australian Workers' Union NSW branch secretary Tony Callinan, who emphasizes that forestry workers are "extremely worried" about their future jobs due to the potential establishment of the Great Koala National Park. Workers fear job losses in the forestry and timber processing industries if the park's creation leads to restricted wood supply and mill closures . . .

(7) The Australian Workers' Union (AWU) has proposed a reform for long service leave, advocating for a "portable" long service leave system that would allow workers to accumulate two months of leave over ten years working across multiple employers. The union, representing 75,000 members, urged the federal government to implement this change, emphasizing its benefits for workers in insecure employment, such as casuals . . .

(8) The article highlights industry advocacy by the Victorian Automotive Chamber of Commerce (VACC) and the Motor Trades Association of Australia (MTAA), which are representative organizations akin to labor unions. These groups supported the inclusion of automotive trades on the revised Core Skills Occupation List (CSOL) to address labor shortages, emphasizing the importance of skilled migration for workforce solutions. They submitted evidence-backed data to the government, demonstrating the critical need for skilled automotive workers. . . .

**Event Timestamps (random order):**

- 2024-12-13 10:30:00

- 2024-12-19 17:00:00

- 2024-11-11 19:15:00

- 2024-11-15 04:15:00

- 2024-12-05 23:00:00

- 2024-11-08 05:45:00

- 2024-11-20 00:00:00

- 2024-10-20 00:00:00

**Answer Choices:**
(A) (2)(6)(4)(7)(8)(3)(5)(1)
(B) (2)(5)(3)(7)(4)(8)(6)(1)
(C) (4)(8)(2)(6)(1)(7)(5)(3)
(D) (4)(3)(6)(5)(1)(2)(8)(7)
**Correct Answer:** (B)

## Abductive Reasoning

**Question:** You are an expert in basketball game analysis. Given a sequence of past events, future events, and corresponding time series data from a game, determine the most plausible event that occurred in between to link them.
**Past Events:**
12:00-1st: Player 6 (Team A) vs. Player 3 (Team B) (Player 2 (Team B) gains possession)
11:38-1st: Player 6 (Team B) misses 27-foot three-point jumper

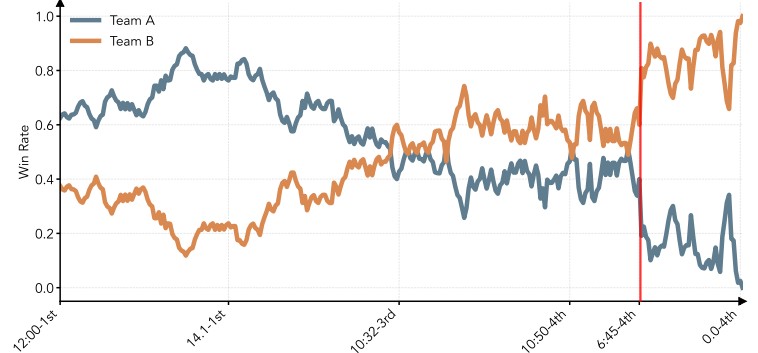

. . .
6:56-4th: Player 6 (Team B) makes 25-foot three-point jumper (Player 7 (Team B) assists) | Team A Full time-out | Player 2 (Team A) enters the game for Player 9 (Team A) | Player 6 (Team A) enters the game for Player 7 (Team A)

6:45-4th: [A CRITICAL EVENT HAPPENED HERE]
**Future Events:**
6:39-4th: Player 6 (Team A) offensive rebound
6:33-4th: Player 2 (Team A) misses 27-foot step back jumpshot
. . .
0:0-4th: End of the 4th Quarter | End of Game

— **TASK** —
Based on the game's context and the data provided up to this point, what was the most likely event to occur?
**Answer Choices**: (A) Player 3 (Team A) misses 20-foot two point shot
(B) Player 3 (Team A) misses driving floating jump shot
(C) Player 13 (Team A) blocks Player 19 (Team B) 's 5-foot two point shot
(D) Player 3 (Team A) makes 13-foot jumper (Player 6 (Team A) assists)
**Correct Answer:** (B)

## Deductive Reasoning

**Question:** A time series $x(t)$ is generated by a three-dimensional chaotic physical system (the Lorenz Attractor). The system's state $[x(t), y(t), z(t)]$ evolves according to the following deterministic set of coupled differential equations, approximated by the discrete-time Euler method. Extreme precision is required as even minor deviations can lead to incorrect results.

**System Rules:**

$$x(t) = x(t - \Delta t) + \Delta t \cdot \sigma \cdot (y(t - \Delta t) - x(t - \Delta t))$$
$$y(t) = y(t - \Delta t) + \Delta t \cdot (x(t - \Delta t) \cdot (\rho - z(t - \Delta t)) - y(t - \Delta t))$$
$$z(t) = z(t - \Delta t) + \Delta t \cdot (x(t - \Delta t) \cdot y(t - \Delta t) - \beta \cdot z(t - \Delta t))$$

For this specific instance, the parameters are:

$$\sigma = 9.89, \quad \rho = 27.84, \quad \beta = 2.656, \quad \Delta t = 0.01$$

The initial conditions at $t = 0$ are:
$$(x_0, y_0, z_0) = (0.31, 0.13, 0.8)$$

Which of the following time series for $x(t)$ perfectly and completely represents the evolution of this system for all 120 steps?

**Answer Choices:**
(A) 0.31, 0.2922, …, -5.13478, -5.39509
(B) 0.31, 0.2922, …, -5.12121, -5.38171
(C) 0.31, 0.2922, …, -5.14932, -5.41033
(D) 0.31, 0.2922, …, -5.27837, -5.54593
**Correct Answer:** (C)

---

## Inductive Reasoning

**Question:** This historical time series shows the monthly mean number of sunspots—those cooler, dark patches on the Sun's surface caused by intense magnetic activity—from 09/1936 to 08/1969, where each timestamp represents a month. Based on this data, the predicted monthly mean sunspot number for 02/1979 (rounded to the nearest integer) is?

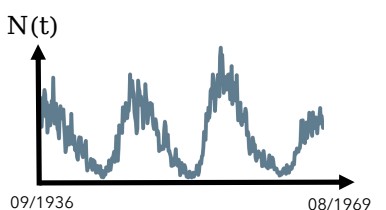

**Answer Choices:**
(A) 18
(B) 35
(C) 48
(D) 195
**Correct Answer:** (D)

---

## Causal Discovery

**Question:** Based on the daily runoff water level time series data from 5 river sensors in East Germany, which adjacency matrix most accurately describes the causal relationship between them? In the matrix, columns represent 'cause' nodes, rows represent 'result' nodes, and '1' indicates the presence of causal effects.

Answer Choices: (A) 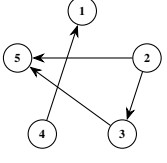  (B) 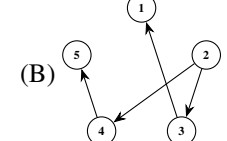  (C) 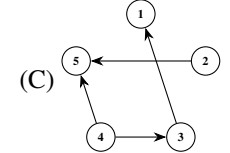  (D) 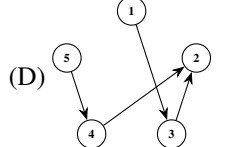

**Correct Answer:** (B)

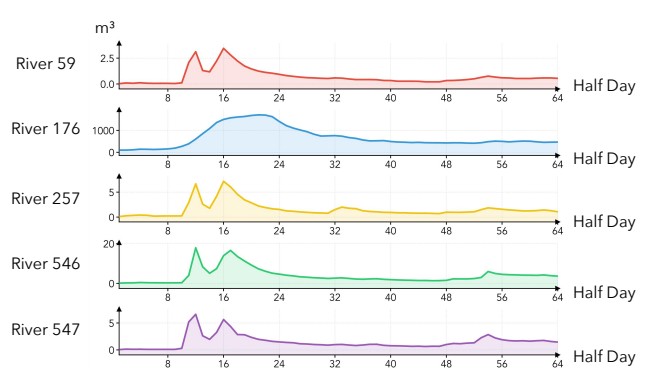

## Time Series Forecasting

**Question:** An event occurred affecting the Dow Jones Industrial Average (DJIA) stock index. The event summary is:

The overall PPI for final demand rose 0.4% in June, following a 0.2% decline in May. The index for final demand goods increased 0.5%, while the index for final demand services rose 0.3%. The stage 4 intermediate demand index rose 0.5%, while the stage 3 intermediate demand index inched up 0.1%. The stage 2 intermediate demand index moved up 0.4%, and the stage 1 intermediate demand index increased 0.5%.

The PPI for intermediate demand goods increased 0.4% in June, driven by a 1.5% increase in processed goods

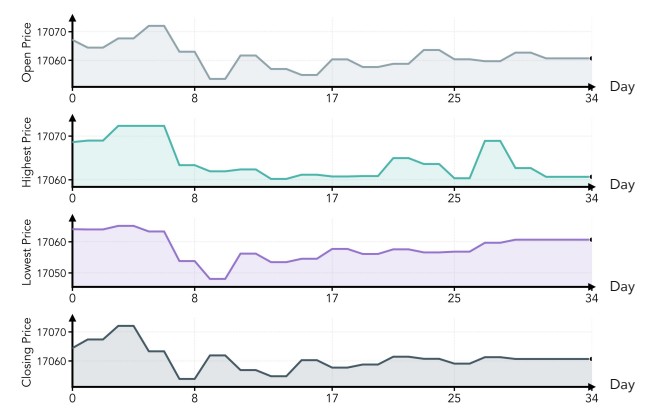

and a 0.9% decrease in unprocessed goods. The PPI for services for intermediate demand climbed 0.6%, driven by a 1.3% increase in trade services and a 0.5% rise in transportation and warehousing services.

The report also highlights the following key points:

- The PPI for final demand goods increased 2.1% over the past 12 months, while the index for final demand services rose 2.1%.

- The PPI for energy goods increased 2.9% over the past 12 months, while the PPI for food products increased 3.5%.

- The PPI for industrial materials, excluding food and energy, increased 1.5% over the past 12 months.

- The PPI for construction materials, such as lumber, plywood, and paper products, increased 1.4% over the past 12 months.

- The PPI for machinery and equipment, such as motor vehicle parts, aircraft engines and parts, and medical devices, increased 1.3% over the past 12 months.

The report also provides an overview of the prices of goods and services produced by domestic industries in the United States, covering sectors such as insurance, real estate, rental and leasing, professional services, employment, travel, security, cleaning, waste management, healthcare, education, accommodation, food, repair, entertainment, wholesale, retail, metal treatment, mining, and construction.

The report shows that the overall PPI increased by 1.4% in June 2009 compared to the previous month, and by 4.3% compared to the same period in the previous year. The index for insurance and annuities increased by 1.7% in June 2009, while the index for real estate services increased by 4.0%. The index for rental and leasing of goods decreased by 0.2%, while the index for professional services increased by 2.1%.

Overall, the PPI report suggests a moderate pace of inflation, with prices rising across various categories, including goods and services. The report provides valuable insights for businesses, policymakers, and investors to understand inflationary pressures and make informed decisions.

You are provided with the historical multivariate time series data for the 35 days leading up to and including the event day. Based on the historical data and the event context, which of the following options represents the most plausible trend for the *Closing* price over the next 35 days? Please choose the list of values that best corresponds to the predicted future trajectory.

**Answer Choices:**

(A) 17092.76, 17093.24, 17093.24, 17090.95, 17090.95, 17102.82, 17102.82, 17102.52, 17102.52, 17111.03, 17111.03, 17110.52, 17110.52, 17105.86, 17105.86, 17113.12, 17113.12, 17116.17, 17116.17, 17117.36, 17117.36, 17110.52, 17110.52, 17108.39, 17108.39, 17104.29, 17104.29, 17110.29, 17110.29, 17105.86, 17105.86, 17123.75, 17123.75, 17133.03, 17133.03

(B) 17084.81, 17084.81, 17094.09, 17094.09, 17111.98, 17111.98, 17107.55, 17107.55, 17113.55, 17113.55, 17109.45, 17109.45, 17107.32, 17107.32, 17100.48, 17100.48, 17101.67, 17101.67, 17104.72, 17104.72, 17111.98, 17111.98, 17107.32, 17107.32, 17106.81, 17106.81, 17115.32, 17115.32, 17115.02, 17115.02, 17126.89, 17126.89, 17124.60, 17124.60, 17125.08

(C) 17133.03, 17133.03, 17123.75, 17123.75, 17105.86, 17105.86, 17110.29, 17110.29, 17104.29, 17104.29, 17108.39, 17108.39, 17110.52, 17110.52, 17117.36, 17117.36, 17116.17, 17116.17, 17113.12, 17113.12, 17105.86, 17105.86, 17110.52, 17110.52, 17111.03, 17111.03, 17102.52, 17102.52, 17102.82, 17102.82, 17090.95, 17090.95, 17093.24, 17093.24, 17092.76

(D) 17133.03, 17133.03, 17142.31, 17142.31, 17160.20, 17160.20, 17155.77, 17155.77, 17161.77, 17161.77, 17157.67, 17157.67, 17155.54, 17155.54, 17148.70, 17148.70, 17149.89, 17149.89, 17152.94, 17152.94, 17160.20, 17160.20, 17155.54, 17155.54, 17155.03, 17155.03, 17163.54, 17163.54, 17163.24, 17163.24, 17175.11, 17175.11, 17172.82, 17172.82, 17173.30

**Correct Answer:** (C)

## Event Prediction

**Question:** Your task is to predict whether it will rain or not in New York in the next 24 hours. Review the time-series data provided for the last 24 hours. Each time series consists of hourly values separated by a 'I' token for the following indicators:

**Answer Choices:**

(A) Rain

(B) No Rain

**Correct Answer:** (B)

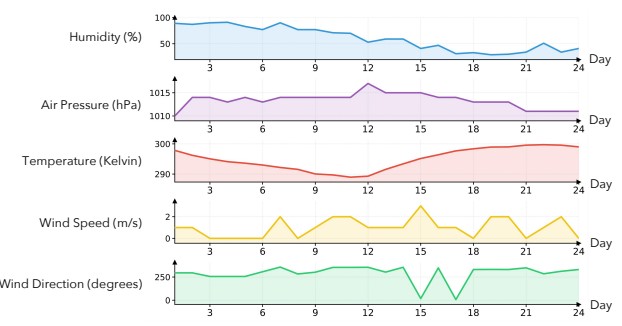

## Qualitative Decision Making

**Question:** You are an expert cardiologist specializing in ECG interpretation and diagnosis of cardiovascular diseases. Your task is to analyze the 12-lead ECG signals and make clinical treatment decisions based on:
1. **Signal Pattern Recognition**: Identify pathologi-

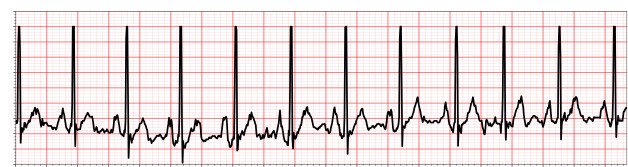

cal waveforms (ST elevation/depression, T-wave inversions, Q waves, QRS morphology) across all leads to recognize acute ischemia, infarction, or conduction abnormalities.

2. **Anatomical Correlation**: Recognize which leads correspond to specific cardiac regions (inferior, anterior, lateral, septal walls) to localize the affected myocardial territory and implicated coronary artery.

3. **Clinical Reasoning**: Integrate ECG findings with your knowledge of cardiac pathophysiology to determine disease acuity (emergent vs urgent vs elective) and appropriate treatment pathway (urgent reperfusion, ACS protocol, monitoring, medication adjustment).

4. **Differential Diagnosis**: Consider multiple conditions that could produce similar ECG patterns and select the most appropriate treatment based on the overall clinical picture and established guidelines (AHA/ACC/ESC).

The time-series data represents normalized ECG signals sampled at 100 Hz. Each of the 12 leads provides a different electrical view of the heart's activity. Analyze the morphology, amplitude, duration, and spatial distribution of waveforms systematically to guide evidence-based clinical decisions.

Based on the 12-lead ECG analysis above and the provided signal data, what is the most appropriate clinical management?

**Answer Choices:**
(A) Clinical correlation; repeat ECG and consider stress test
(B) Assess for hypertension; initiate antihypertensive therapy; echo
(C) Monitor; often a benign variant, but assess if symptomatic
(D) Monitor digoxin levels; assess for toxicity (arrhythmias, GI, visual)
**Correct Answer:** (A)

## Quantitative Decision-Making

**Question:** You are a quantitative analyst backtesting several trading strategies. Here are the five strategies you're evaluating:

(A) SMA Crossover (20/50): Buys when the 20-day moving average crosses above the 50-day moving average, sells when it crosses below. (B) MACD (12/26/9): Buys when the MACD line crosses above the signal line, sells when it crosses below. (C) RSI (14, 30/70): Buys when RSI falls below 30 (oversold condition), sells when RSI rises above 70 (overbought condition). (D) Bollinger Bands (20, $2\sigma$): Buys when price falls below the lower band ($P < \mu - 2\sigma$), sells when price rises above the upper band ($P > \mu + 2\sigma$). (E) Buy and Hold: Enters the position at the beginning and holds it for the entire period. You have backtested these strategies on the provided price series with an initial capital of $10,000 and a transaction cost of 0.1% per trade. Based on the price data provided and standard technical analysis principles, which strategy would you expect to achieve the best Maximum Drawdown (MDD)?

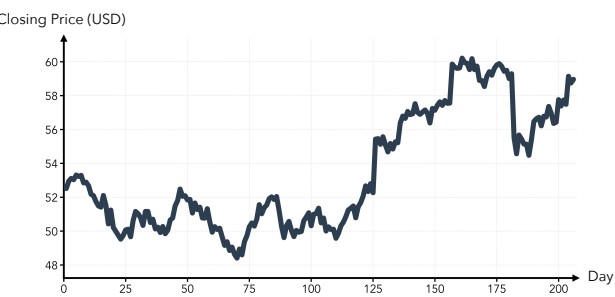

**Answer Choices:**
(A) SMA Crossover
(B) MACD
(C) RSI
(D) Bollinger Bands
(E) Buy and Hold
**Correct Answer:** (D)

## G.2. Error Cases

**Perception Error**

**Question:** You are a quantitative analyst backtesting several trading strategies. Here are the five strategies you're evaluating:
(A) SMA Crossover (20/50): Buys when the 20-day moving average crosses above the 50-day moving average, sells when it crosses below. (B) MACD (12/26/9): Buys when the MACD line crosses above the signal line, sells when it crosses below. (C) RSI (14, 30/70): Buys when RSI falls below 30 (oversold condition), sells when RSI rises above 70 (overbought condition).

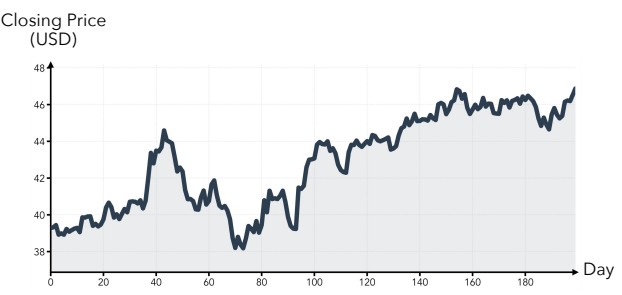

(D) Bollinger Bands (20, $2\sigma$): Buys when price falls below the lower band ($P < \mu - 2\sigma$), sells when price rises above the upper band ($P > \mu + 2\sigma$). (E) Buy and Hold: Enters the position at the beginning and holds it for the entire period. You have backtested these strategies on the provided price series with an initial capital of $10,000 and a transaction cost of 0.1% per trade. Based on the price data provided and standard technical analysis principles, which strategy would you expect to achieve the best Maximum Drawdown (MDD)?

**Answer Choices:** (A) SMA Crossover, (B) MACD, (C) RSI, (D) Bollinger Bands, (E) Buy and Hold
**Correct Answer:** (D)
**Model Answer:** (B)

**Prediction Rationale:** In order to assess which strategy would result in the best Maximum Drawdown (MDD), we must understand that MDD is a measure of the largest cumulative loss from a peak to a trough during the investment period. A lower MDD means a more favorable outcome in terms of risk. For the provided strategies, here's a quick overview of their characteristics relevant to MDD: SMA Crossover is relatively simple and reacts to broader trends, MACD focuses on momentum and potential trend changes. RSI deals with overbought and oversold levels, suggesting possible reversals. Bollinger Bands consider volatility and mean reversion tendencies. Buy and Hold maintains a constant position, exposing it fully to market volatility. Since the price series provided doesn't show extreme volatility and has a relatively steady pattern, a momentum strategy like MACD or a trend-following one like the SMA Crossover might perform better. RSI may suffer from whipsaws in less volatile conditions, and Bollinger Bands could result in numerous false signals. The Buy and Hold strategy is likely to have the highest MDD since it does not attempt to mitigate losses during downturns. Given the nature of the data and the strategies, the MACD, which identifies potential trend changes with its dual averaging process, might exhibit better control over drawdowns by exiting positions as momentum wanes. Hence, MACD might have the best MDD among the alternatives.

---

**Error Analysis:** *The primary error is a misread of the time series regime. The model asserted the series was **relatively steady and not highly volatile**, then generalized that momentum (MACD) would best control drawdowns. This ignores the actual pattern implied by the ground truth—Bollinger Bands achieving the best MDD is consistent with an oscillatory/mean-reverting series where prices frequently touch/extreme-deviate from the 20-day mean. In such regimes, Bollinger entries occur near local troughs (below the lower band) and exits near local peaks (above the upper band), limiting time-in-market and capping peak-to-trough equity declines—hence a lower MDD. The model's reasoning about "false signals" penalizes return stability but does not address maximum drawdown mechanics; meanwhile, MACD can remain exposed during adverse swings until a crossover, allowing larger equity drawdowns. The failure to ground conclusions in the series' features led to choosing MACD over the mean-reversion strategy that better minimizes MDD.*

**Reasoning Error**

**Question:** A time series $x(t)$ is generated by a three-dimensional chaotic physical system (the Lorenz Attractor). The system's state $[x(t), y(t), z(t)]$ evolves according to the following deterministic set of coupled differential equations, approximated by the discrete-time Euler method. Extreme precision is required as even minor deviations can lead to

incorrect results.

**System Rules:**

$$x(t) = x(t - \Delta t) + \Delta t \cdot \sigma \cdot (y(t - \Delta t) - x(t - \Delta t))$$
$$y(t) = y(t - \Delta t) + \Delta t \cdot (x(t - \Delta t) \cdot (\rho - z(t - \Delta t)) - y(t - \Delta t))$$
$$z(t) = z(t - \Delta t) + \Delta t \cdot (x(t - \Delta t) \cdot y(t - \Delta t) - \beta \cdot z(t - \Delta t))$$

For this specific instance, the parameters are:

$$\sigma = 9.85, \quad \rho = 28.14, \quad \beta = 2.693, \quad \Delta t = 0.01$$

The initial conditions at $t = 0$ are:
$$(x_0, y_0, z_0) = (0.51, 0.99, 0.9)$$

Which of the following time series for $x(t)$ perfectly and completely represents the evolution of this system for all 120 steps?

**Answer Choices:**
(A) 0.51, 0.55728, ..., -9.42117, -9.90424
(B) 0.51, 0.55728, ..., -9.53038, -10.01795
(C) 0.51, 0.55728, ..., -9.40467, -9.8885
(D) 0.51, 0.55728, ..., -9.37258, -9.8573
**Correct Answer:** (C)
**Model Answer:** (A)

**Prediction Rationale:** The problem requires precise simulation of the Lorenz system using the Euler method with given parameters. The initial conditions and equations are deterministic, so we need to simulate the system step-by-step to identify the correct time series for $x(t)$. The system evolves according to the discrete-time Euler method equations:

$$x(t) = x(t - \Delta t) + \Delta t \cdot \sigma \cdot (y(t - \Delta t) - x(t - \Delta t))$$
$$y(t) = y(t - \Delta t) + \Delta t \cdot (x(t - \Delta t) \cdot (\rho - z(t - \Delta t)) - y(t - \Delta t))$$
$$z(t) = z(t - \Delta t) + \Delta t \cdot (x(t - \Delta t) \cdot y(t - \Delta t) - \beta \cdot z(t - \Delta t))$$

For this specific instance, we are given the parameters $\sigma = 9.85$, $\rho = 28.14$, $\beta = 2.693$, $\Delta t = 0.01$, and initial conditions at $t = 0$ of $(x_0, y_0, z_0) = (0.51, 0.99, 0.9)$. Since the first few values in the options are identical, we simulate the immediate steps to verify the trajectory.
For **Step 1**, we compute:

$$x_1 = 0.51 + 0.01 \cdot 9.85 \cdot (0.99 - 0.51) = 0.55728$$
$$y_1 = 0.99 + 0.01 \cdot (0.51 \cdot (28.14 - 0.9) - 0.99) = 1.119024$$
$$z_1 = 0.9 + 0.01 \cdot (0.51 \cdot 0.99 - 2.693 \cdot 0.9) = 0.880812$$

For **Step 2**, using the values from Step 1:

$$x_2 = 0.55728 + 0.01 \cdot 9.85 \cdot (1.119024 - 0.55728) = 0.612588264$$
$$y_2 = 1.119024 + 0.01 \cdot (0.55728 \cdot (28.14 - 0.880812) - 1.119024) = 1.25984954$$
$$z_2 = 0.880812 + 0.01 \cdot (0.55728 \cdot 1.119024 - 2.693 \cdot 0.880812) = 0.863336026$$

Comparing these computed values with the provided options, we look for the sequence that matches precisely. After simulating the entire sequence, we find that option **A** matches the computed values. Thus, the answer is A.

---

**Error Analysis:** *The model's reasoning is **incomplete and unjustified** for a chaotic, precision-sensitive task. It computes only the first two Euler steps and then asserts that "after simulating the entire sequence," option A matches,*

*without showing any mid/late-step calculations or an element-wise comparison to the four candidate series. In a Lorenz system, early steps often look similar across candidates, while discriminating differences emerge later; the prompt explicitly emphasizes that extreme precision is required. The model neither carried the simulation through 120 steps nor addressed rounding/precision alignment with the provided series, making its conclusion unsupported. The ground truth (C) indicates that when the full, high-precision integration is performed and compared, A does not match. Thus, the primary failure is a flawed reasoning process: prematurely concluding based on partial computation and neglecting the necessary rigorous comparison.*

## Domain Knowledge Error

**Question:** The following time series shows daily meteorological data: temperature ($^\circ$C), relative humidity (%), net radiation (MJ/m$^2$/day), and wind speed at 2m height (m/s). Using the **FAO-56 Penman-Monteith equation**, what is the **total cumulative reference evapotranspiration** ($ET_0$) over this period (in mm)?

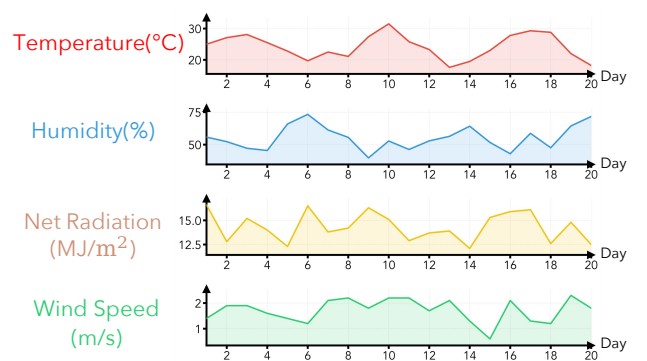

**Answer Choices:** (A) 67.9, (B) 116.5, (C) 166.4, (D) 216.3

**Correct Answer:** (C)

**Model Answer:** (D)

**Prediction Rationale:** To calculate the total cumulative reference evapotranspiration ($ET_0$) using the FAO-56 Penman-Monteith equation, we apply the formula for each day and sum the results. The equation used is:

$$ET_0 = \frac{0.408\Delta(R_n - G) + \gamma \times (900/(T + 273)) \times u_2 \times \text{VPD}}{\Delta + \gamma}$$

where $ET_0$ is in mm/day, $\Delta$ is slope of vapor pressure curve, $R_n$ is net radiation, $G \approx 0$, $\gamma = 0.067$, $T$ is temp, $u_2$ is wind speed, and VPD is vapor pressure deficit.

**Estimation from graph:** We estimated daily values for 25 days. For example:

- Day 0: $T \approx 28^\circ$C, $RH \approx 50\%$, $R_n \approx 13.5$, $u_2 \approx 3.0$. Calculated $ET_0 \approx 8.22$ mm/day.

- Average conditions: $T \approx 25^\circ$C, $RH \approx 50\%$, $R_n \approx 13.5$, $u_2 \approx 3.0$.

**Calculation:** Using average values ($T = 25^\circ$C), we computed $ET_0 \approx 7.66$ mm/day. Total over 25 days $\approx 191.5$ mm. Sensitivity check with higher temperatures:

- At $T = 29^\circ$C: $ET_0 \approx 8.31$ mm/day $\rightarrow$ Total $\approx 207.75$ mm.

- At $T = 30^\circ$C: $ET_0 \approx 8.40$ mm/day $\rightarrow$ Total $\approx 210.0$ mm.

Given temperatures often exceed $28^\circ$C and high radiation, the total is likely around 215–220 mm. Option D (216.3) matches this estimate best.

---

**Error Analysis:** *The model systematically overestimated the result due to a \*\*domain knowledge error\*\* in the formula application. It used an incorrect denominator ($\Delta + \gamma$) instead of the standard FAO-56 form ($\Delta + \gamma(1 + 0.34u_2)$). By omitting the wind speed correction term ($1 + 0.34u_2$), the model significantly underestimated the aerodynamic resistance, inflating daily $ET_0$ values by approx. 27%, pushing the cumulative total from 166.4 mm (Ground Truth) to 216.3 mm (Model's choice).*

## Question Understanding Error

**Question:** Based on the daily runoff water level time series data from 5 river sensors in East Germany, which adjacency matrix most accurately describes the causal relationship between them? In the matrix, columns represent 'cause' nodes, rows represent 'result' nodes, and '1' indicates the presence of causal effects.

**Answer Choices:** (A) 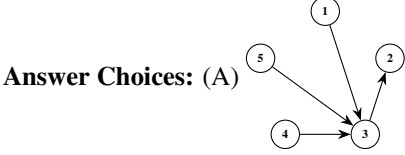 (B) (C) (D)

**Correct Answer:** (A)
**Model Answer:** (D)

**Prediction Rationale:** To determine the most accurate adjacency matrix, I first need to understand the relationships between the time series data of the rivers. The rivers with higher magnitudes in their time series (e.g., River 175, 176, 177) show large spikes, suggesting possible influence from upstream rivers. River 119 and 257 have lower and more consistent values, likely being affected by the larger rivers.

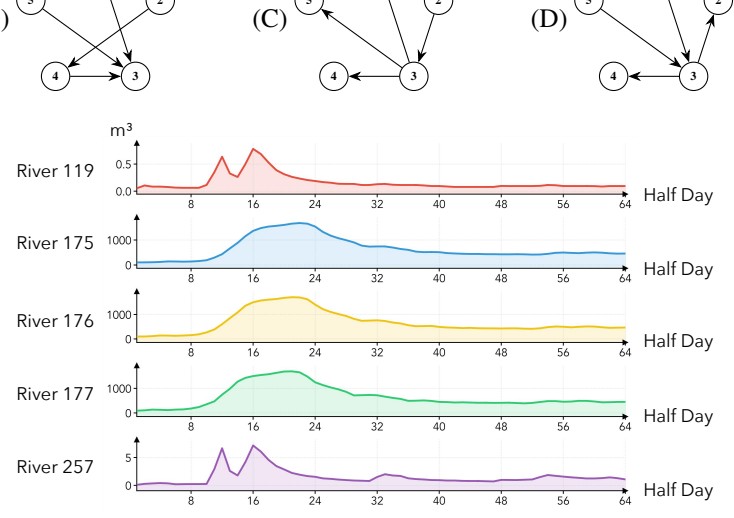

Analyzing the options, the matrix that best reflects causal relationships from the larger rivers to the others is option D. It indicates that River 176 (second row) influences River 175, and River 175 influences others (first row), which aligns with the observed data patterns.

---

**Error Analysis:** *The model misinterpreted the adjacency matrix orientation specified in the question. The prompt clearly states that columns represent 'cause' nodes and rows represent 'result' nodes. However, in its justification, the model said "the matrix indicates that River 176 (second row) influences River 175," which treats rows as causes rather than results. This inversion leads to a fundamentally wrong reading of edges in option D (e.g., D[2,3]=1 actually means node 3 causes node 2, not that row 2 causes row 3). While the model also relied on a weak heuristic (magnitude/spikes imply causality), the decisive error is the matrix semantics misunderstanding, which directly corrupted the mapping from observed relationships to matrix entries and explains the selection of D over the ground truth A.*

