# OpenReview forum: "TSRBench: A Comprehensive Multi-task Multi-modal Time Series Reasoning Benchmark for Generalist Models"
_ICML.cc/2026/Conference — ICML 2026 regular_

### Official Review · Reviewer_eBxP · 2026-03-09

**Soundness:** 2
**Presentation:** 3
**Significance:** 3
**Originality:** 2
**Overall Recommendation:** 4
**Confidence:** 4

**Summary:**

This paper introduces TSRBench, a new benchmark for time series reasoning that covers four major categories and 15 tasks. The benchmark contains around 4k samples, which is a reasonable scale for this setting. Overall, the paper is clearly written and easy to follow, and it presents several insightful findings.

**Compliance With Llm Reviewing Policy:**

Affirmed.

**Final Justification:**

I appreciate the authors' effort in preparing the rebuttal, which addressed some of my questions and concerns. I have raised my score to 4 considering both the rebuttal and the overall quality of the submission.

**Key Questions For Authors:**

- Table 1 claims that TSRBench is multivariate. (1) In a single QA instance, what is the maximum number of variables involved?  (2) What proportion of the benchmark consists of multivariate QA instances? Based on Figure 1, it seems that only Event Prediction involves more than two variables, while all the remaining tasks involve at most two variables. Please clarify this in much greater detail, ideally with a summary table broken down by all of the 15 tasks, so as to comprehensively justify that the claim of TSRBench being multivariate is both reasonable and fair.

- One important aspect that distinguishes TSRBench from previous benchmarks is its claim of being multimodal, namely involving both text and vision modalities. However, many important details about the visualization design are not discussed sufficiently. For example: (1) whether legends are included, (2) how line charts are standardized across tasks, (3) whether grid lines should be added to help VLMs better estimate numerical values, (4) whether marking turning points or abrupt changes could improve performance, and (5) whether task-specific visual backgrounds should be preserved (for example, in 12-lead ECG data, whether retaining the background grid could help models make better decisions). These issues are all relevant, but they are not discussed in enough detail.

**Limitations:**

- The authors should include a separate section to discuss the limitations of the current work in a fair and transparent manner. The current version does not provide the discussion.

**Strengths And Weaknesses:**

**Strengths**

- The benchmark draws from diverse data sources, with synthetic data and real-world web data complementing each other well. In addition to time series reasoning tasks, it also includes temporal reasoning tasks, such as Temporal Relation Reasoning. The figures and tables are of good quality.

**Weaknesses**

- Overall the 4 findings are superficial. They are all based on analyses of the results in Table 2, or results directly transformed from Table 2. By contrast, some of the more important aspects of the paper, such as how the images are constructed and why general-purpose large models fail, are not analyzed in enough detail and do not provide sufficient insights.
- In Finding 2, the last sentence states “See Appendix F.1 for correlations within tasks.” However, Table 8 in Appendix F.1 does not correspond to Finding 2; instead, it appears to present additional experimental results for Finding 1.
- Finding 3 claims that “textual and visual representations ... are strongly complementary.” However, this complementarity is not sufficiently supported by the experiments in the paper. How exactly is this complementarity established? E.g., does combining the two modalities help the model solve questions that cannot be solved with either modality alone? From the figure, however, the conclusion seems to be the opposite: using both modalities jointly actually leads to worse performance. The experiments for Finding 3 can only show that textual and visual representations have substantial overlap, rather than demonstrating complementarity.
- Finding 4 groups the tasks into two categories: high-variance and low-variance. However, Finding 4 does not include any test-time adaptation or post-training experiments for either category. For this reason, I am not convinced by Finding 4.
- The tool-use setting in Section 4.4 may be biased, because it is difficult to control whether proprietary models such as o4-mini, GPT-5-mini, and GPT-5 rely on internal tool use during reasoning, for example by implicitly executing Python code. As a result, the conclusions drawn from Table 4 are not reliable.
- In fact, the error analysis in Section 4.5 is a very valuable experiment, since it could help researchers better understand why general-purpose LLMs or VLMs fail, and thereby provide useful guidance for improving TSLMs. However, the authors’ analysis of Figure 9 is rather superficial. In particular, two points need clarification: (1) What are the clear boundaries among the four error types, namely Reasoning, Perception, Question Understanding, and Domain Knowledge? (2) How are these error types determined in practice? For example, are they identified by human annotators or by an LLMs-as-judgement protocol?

---

> ### Author Rebuttal · Authors · 2026-03-30
>
> Thanks for your detailed comments.
> - Q1: How are images constructed? Why do general-purpose large models fail?
>
> A1: Time series are rendered as line charts in vertically stacked subplots, with grid lines enabled and series names as axis labels. Large models are primarily trained on natural languag rather than continuous numerical sequences, making it inherently difficult to precisely track values and perform grounded reasoning across hundreds of time points.
>
> - Q2: Appendix F.1 mislabeled.
>
> A2: Thanks for pointing it out. Appendix F.1 presents fine-grained results for Finding 1. We will correct the reference.
>
> - Q3: (1) How is complementarity established? (2) Combining two modalities helps the model solve questions that cannot be solved with either modality alone? (3) Does joint input lead to worse performance? (4) Overlap vs. complementarity.
>
> A3: We respond point-by-point:
>
> (1) We define complementarity as two modalities solving largely non-overlapping subsets of problems. Figure 5 (left) confirms this: the intersection of correct answers from both modalities is low while the union is high.
>
> (2) Yes. As shown in Figure 5 (right), the T+V correct set is not fully covered by the union of individual modality correct sets, confirming that joint input enables models to solve some samples that neither modality solves alone.
>
> (3) No. As shwon in Table 2, using joint input for GPT-5 achieves 55.6% overall accuracy, outperforming text at 55.5% and vision at 52.4%.
>
> (4) The low overlap between modalities is evidenced by Figure 5 (left): the intersection of their correct solution sets yields low accuracy while the union yields high accuracy, confirming that text and vision solve largely non-overlapping subsets of problems.
>
> - Q4: Finding 4 lacks test-time adaptation or post-training experiments.
>
> A4: TSRBench primarily focuses on evaluating the capabilities of generalist models. Finding 4 is a diagnostic observation to guide future methods. Potential methods include on-policy distillation or synthesizing more data for pre-training. We will discuss in Appendix B Future Work.
>
> - Q5: Tool-use setting may be biased due to proprietary models' internal tools.
>
> A5: Our tool augmentation does not invoke model-internal capabilities. We extract statistical features using external tools and inject them into the input prompt, ensuring all models receive identical augmented information.
>
> - Q6: Clarifications on (1) error type boundaries (2) How are these error types determined?
>
> A6: (1) Question Understanding: model misinterprets the question; Perception: model misreads the time series; Domain Knowledge: wrong domain rules applied; Reasoning: correct observations but flawed logical inference. (2) All error categorizations are performed by human.
>
> - Q7: (1) Maximum number of variables? (2) Prohumanson of the benchmark consists of multivariate QA instances? Provide a summary table.
>
> A7: (1) The maximum number of variables is 12.
> (2) We analyze the proportion of multivariate instances across tasks:
> |PR|NU|AD|SA|ER|CD|AR|TR|NR|DR|IR|TSF|EP|QualDM|QuantDM|Overall|
> |-|-|-|-|-|-|-|-|-|-|-|-|-|-|-|-|
> |3.8%|14.9%|31.0%|100%|57.1%|100%|0%|0%|62.5%|0%|50.0%|100%|100%|100%|0%|58.1%|
>
> In summary, 58.1% instances are multivariate; we will include this table in the main text.
>
> - Q8: Visualization design details: (1) whether legends are included, (2) how line charts are standardized across tasks, (3) whether grid lines should be added to help VLMs better estimate numerical values, (4)  whether marking turning points improves performance, and (5) Whether task-specific visual backgrounds (e.g., ECG grids) should be preserved.
>
> A8: We respond point-by-point:
>
> (1) Each time series has a legend
>
> (2) All tasks share identical plotting parameters with the identical width, and the height depends on the number of time series
>
> (3) Yes, grid lines are always enabled
>
> (4) We investigate the impact of adding a turning point by marking the peak as red and valley as green. We report the performance change on 3 VLMs:
> |Δ|PR|NU|AD|SA|ER|CD|AR|TR|NR|DR|IR|TSF|EP|QualDM|QuantDM|Overall|
> |-|-|-|-|-|-|-|-|-|-|-|-|-|-|-|-|-|
> |Qwen2.5-VL-7B|+1.1|−6.9|+1.2|+8.8|+7.0|−1.7|−24.4|−2.4|−1.6|−2.3|−4.6|+0.4|−9.6|+3.6|−4.3|−1.5|
> |InternVL3.5-8B|+0.3|−10.4|−4.3|−4.4|+0.3|−6.6|−3.3|−1.2|+5.8|+2.4|−4.0|−0.4|−15.0|+2.4|+1.4|−1.5|
> |MiMo-VL-7B|−4.1|−16.1|−5.8|−1.6|−5.1|+6.0|−20.0|−7.5|+2.8|−1.6|−3.0|+7.0|−8.9|+6.8|+1.3|−0.8|
>
> We observe that most tasks degrade. These tasks likely do not rely on turning points, marking them may introduce noise that hinders time series understanding.
>
> (5) We evaluate 3 VLMs with ECG background grids on QualDM:
> |Model|default|+background grid|Δ|
> |-|-|-|-|
> |Qwen2.5-VL-7B|26.3|26.1|-0.2|
> |InternVL3.5-8B|33.4|33.6|+0.2|
> |MiMo-VL-7B|29.7|29.6|-0.1|
>
> Background grids show no obvious improvement, likely because default grid lines already provide sufficient numerical reference.
>
> - Q9: Incorporation of Limitation Section
>
> A9: We will add in the revision.

---

> > ### Author Rebuttal · Reviewer_eBxP · 2026-04-02
> >
> > Thank you and the rebuttal has resolved some of my questions and concerns. I acknowledge that addressing them all in a short rebuttal like this could be challenging, and I am willing to increase my overall rating from 3 to 4 considering both the rebuttal and the potential impact of this work. As an additional comment, the authors are encouraged to benchmark more up-to-date time series language models such as OpenTSLM and TimeOmni.

---

> > > ### Author Response · Authors · 2026-04-03
> > >
> > > Thank you so much for your encouraging and positive feedback!
> > >
> > > To address your additional comments, we follow your suggestions to evaluate 2 OpenTSLM [1] variants trained for time series QA and TimeOmni-1 [2] on our benchmark and report the performance:
> > >
> > > | Model | PR | NU | AD | SA | ER | CD | AR | TR | NR | DR | IR | TSF | EP | QualDM | QuantDM | Overall |
> > > | :--- | :--- | :--- | :--- | :--- | :--- | :--- | :--- | :--- | :--- | :--- | :--- | :--- | :--- | :--- | :--- | :--- |
> > > | OpenTSLM-3B-tsqa-SP | 39.9 | 40.2 | 35.7 | 43.4 | 20.3 | 32.3 | 33.3 | 28.1 | 24.5 | 26.0 | 32.0 | 32.5 | 35.6 | 28.4 | 28.7 | 31.0 |
> > > | OpenTSLM-3B-tsqa-Flamingo | 41.2 | 41.4 | 34.1 | 38.9 | 22.9 | 30.3 | 36.0 | 21.2 | 22.5 | 28.8 | 29.0 | 28.1 | 45.6 | 25.4 | 29.7 | 30.7 |
> > > | TimeOmni-1-7B | 55.0 | 59.8 | 41.9 | 64.6 | 28.0 | 35.3 | 46.7 | 24.4 | 31.5 | 22.8 | 34.0 | 30.0 | 49.4 | 34.0 | 30.3 | 36.7 |
> > >
> > > The results show that TimeOmni-1-7B achieves 36.7% overall accuracy, a 3.0% improvement over its base model Qwen2.5-7B-Instruct (33.7%), indicating post-training LLMs yields meaningful gains on time series reasoning tasks. OpenTSLM variants (based on Llama-3.2-3B) underperform similarly-sized Qwen2.5-3B (33.2%),  possibly due to differences in base model capabilities. We will make sure to incorporate these results into Table 2 in the main text.
> > >
> > > We sincerely appreciate your valuable feedback again!
> > >
> > > References:
> > >
> > > [1] Langer, Patrick, et al. "Opentslm: Time-series language models for reasoning over multivariate medical text-and time-series data." arXiv preprint arXiv:2510.02410 (2025).
> > >
> > > [2] Tong Guan, et al. "TimeOmni-1: Incentivizing Complex Reasoning with Time Series in Large Language Models." The Fourteenth International Conference on Learning Representations

---

### Official Review · Reviewer_KtGn · 2026-03-09

**Soundness:** 3
**Presentation:** 3
**Significance:** 2
**Originality:** 3
**Overall Recommendation:** 4
**Confidence:** 4

**Summary:**

This paper introduces TSRBENCH, a comprehensive multi-modal benchmark featuring 4,125 problems across 15 tasks to evaluate generalist models on time series perception, reasoning, prediction, and decision-making, ultimately revealing critical weaknesses in current models' forecasting and cross-modal fusion capabilities.

**Compliance With Llm Reviewing Policy:**

Affirmed.

**Key Questions For Authors:**

No more questions

**Limitations:**

yes

**Strengths And Weaknesses:**

S1. The benchmark pushes the field beyond simple pattern recognition by defining a robust taxonomy of 15 tasks categorized into four distinct cognitive dimensions: Perception, Reasoning, Prediction, and Decision-Making.

S2. The dataset successfully balances real-world complexity through human-verified web data collection with verifiable precision using code-synthesized data , which minimizes ambiguity and noise in the ground truth

S3. The extensive evaluation of language models yields valuable observations for the community. Notably, the authors identify that traditional scaling laws break down specifically for time series forecasting tasks.

W1. The study evaluates an impressive array of models across textual, visual, and embedded modalities. However, it lacks a detailed exploration or ablation of how different prompting techniques impact the reasoning dimensions compared to the baseline zero-shot approach

W2. Simplifying the complex numerical prediction into a multiple-choice format artificially lowers the difficulty. This may disproportionately benefit smaller LLMs, potentially explaining the observed breakdown of scaling laws in prediction tasks rather than reflecting a true limitation of larger models.

(S for Strengths, and W for Weaknesses)

---

> ### Author Rebuttal · Authors · 2026-03-30
>
> We thank the reviewer for the positive feedback and recognize our evaluation provides **valuable insights**. For the concerns, we detail our responses as follows:
>
> - Q1: Lacks a detailed exploration or ablation of how different prompting techniques impact the reasoning dimensions compared to the baseline zero-shot approach
>
> A1: We follow your suggestion to conduct additional experiments on 1-shot prompting with 3 LLMs and VLMs, and the performance changes compared to zero-shot are shown below:
>
> | Δ | PR | NU | AD | SA | ER | CD | AR | TR | NR | DR | IR | TSF | EP | QualDM | QuantDM | Overall |
> |---|---|---|---|---|---|---|---|---|---|---|---|---|---|---|---|---|
> | Qwen2.5-7B | +0.5 | +0.0 | +3.9 | +6.2 | +4.3 | -2.0 | +0.7 | +0.6 | +0.2 | +2.0 | +0.0 | +0.3 | +0.3 | +3.9 | +2.7 | +1.3 |
> | Gemma3-12B | +0.8 | +1.1 | -1.6 | +0.9 | +2.9 | +0.3 | +4.7 | +0.6 | +6.0 | +0.4 | +0.0 | -1.7 | +1.1 | +0.6 | +0.7 | +1.0 |
> | InternLM3-8B | -2.4 | +9.2 | +7.0 | +5.3 | +0.6 | +0.7 | -2.0 | +0.6 | +0.5 | +1.2 | -2.0 | +0.6 | +8.9 | +0.3 | +0.7 | +1.4 |
> | Qwen2.5-VL-7B | +3.2 | -2.3 | +1.6 | +3.5 | +0.3 | +2.7 | -2.0 | +0.6 | +0.2 | +0.4 | +0.0 | +0.3 | +2.5 | +1.5 | +0.3 | +1.0 |
> | InternVL3.5-8B | +1.1 | -1.1 | +0.8 | +0.9 | +0.9 | -1.0 | -0.7 | +0.6 | +1.0 | -0.4 | +1.0 | +0.8 | +1.1 | +0.9 | +4.7 | +0.9 |
> | MiMo-VL-7B-RL | +1.6 | +1.1 | +3.9 | +1.8 | +3.4 | +2.0 | -0.7 | +1.9 | +2.0 | +1.6 | +1.0 | +2.4 | +5.8 | +3.3 | +4.0 | +2.6 |
>
> The results show that the one-shot prompting strategy improves on most tasks and leads to consistent improvement on overall accuracy compared to zero-shot prompting. We will add a discussion in the main text.
>
> - Q2. Simplifying the complex numerical prediction into a multiple-choice format artificially lowers the difficulty. This may disproportionately benefit smaller LLMs, potentially explaining the observed breakdown of scaling laws in prediction tasks rather than reflecting a true limitation of larger models.
>
> A2: Thank you for your suggestion. Formatting time series forecasting as multiple-choice enables us to analyze across tasks in a holistic approach. To further address your concern, we conduct experiments with LLMs and VLMs on time series forecasting tasks by open-ended numerical forecasting, and use nMAE as the metric. After obtaining the results, we compute the Spearman's rank correlation between LLM and VLM performance and model size, and compare with the MCQ version. The results shown below:
>
> |Spearman| MCQ |Numerical Forecasting |
> | :--- | :--- | :--- |
> | LLMs | -0.55 |  -0.61 |
> | VLMs  | 0.03 |  -0.05 |
>
> The results show that the multi-choice format and open-ended numerical forecasting show high correlation: Open-ended forecasting still shows a negative or orthogonal relation to the model size. This indicates that the multiple-choice accuracy is consistent with actual forecasting quality and does not benefit smaller models.

---

> > ### Author Rebuttal · Reviewer_KtGn · 2026-04-08
> >
> > Thank you for your response and the additional clarifications. However, considering the paper's novelty and the lack of strictly dynamically updated data, I will maintain my original score.

---

> > > ### Author Response · Authors · 2026-04-08
> > >
> > > Thank you for your positive evaluation of our paper. We would like to address your comments further as follows:
> > > ﻿
> > > - **Novelty**: Time series reasoning is a fundamental capability for building intelligent systems and crucial for critical applications such as clinical management, disaster forecasting, and scientific discovery, yet existing benchmarks either focus on narrow time series analysis tasks or restricted domains, leaving a critical gap in holistic evaluation. To this end, our proposed TSRBench covers **4 dimensions and 15 diverse tasks** to evaluates time series reasoning on generalist models comprehensively. After constructing the benchmark, we conduct extensive experiments on more than **30** models, revealing multiple novel findings on current generalist models, such as the breakdown of scaling laws in time series prediction and the strong complementarity yet ineffective fusion of textual and visual time series representations. These findings inspire multiple research directions for the future (as discussed in Appendix D).
> > > ﻿
> > > - **Capability of Dynamically Updated Data**: TSRBench supports dynamic data updates. First, our synthetic data pipeline is inherently dynamic since the code-based generation functions can produce unlimited new instances by varying parameters (e.g., Lorenz system coefficients), enabling dynamic data refreshing. Second, our web-data collection pipeline can be re-run periodically to incorporate new events and time series (e.g., latest news and stock prices), refreshing outdated samples.
> > > ﻿
> > > We hope these clarifications can further address your comments.

---

### Official Review · Reviewer_Cgj8 · 2026-03-10

**Soundness:** 2
**Presentation:** 4
**Significance:** 3
**Originality:** 3
**Overall Recommendation:** 4
**Confidence:** 3

**Summary:**

The paper introduces TSRBENCH, a comprehensive, multi-modal benchmark designed to evaluate generalist LLMs and VLMs on time series reasoning. Comprising 4125 problems across 14 domains, it categorizes tasks into four dimensions: Perception, Reasoning, Prediction, and Decision-Making. By evaluating over 30 models using text, visual, and combined inputs, the authors reveal that current models perform reasonably well on perception but struggle significantly with complex reasoning and forecasting. The paper also highlights an ongoing challenge in effectively fusing multimodal time series inputs.

**Compliance With Llm Reviewing Policy:**

Affirmed.

**Final Justification:**

I recommend Weak Accept. The rebuttal effectively resolved my core concerns about the multiple-choice format's validity and the results' statistical robustness. Despite remaining limitations in benchmark scale and the mixed results of tool augmentation, the paper's 15-task taxonomy offers a well-structured, solid contribution to the field.

**Key Questions For Authors:**

1. The paper reformulates time-series forecasting into a multiple-choice format, which standardizes the evaluation interface but also changes the nature of the original task. Can the authors clarify how performance under this setup relates to real open-ended numerical forecasting? Is there any evidence that multiple-choice accuracy is consistent with, or predictive of, actual forecasting quality?
2. The paper currently provides limited statistical support for fine-grained score differences. Given the many small gaps in Table 2, could the authors report confidence intervals or error bars to show whether these differences are robust?
3. Table 4 shows only small overall improvements from tool augmentation, with degradations on several dimensions. If the benefits of tool augmentation are limited to only a small subset of tasks, how would the authors more precisely delimit its scope of usefulness and characterize its practical value?

**Limitations:**

No.
The paper does not yet sufficiently discuss several important limitations. Formulating the Prediction dimension as a multiple-choice evaluation standardizes the interface, but it still differs substantially from real open-ended forecasting. With only 4.1K questions in total, the benchmark is better suited for broad taxonomy coverage than for highly stable fine-grained leaderboard comparisons. And although the paper emphasizes multimodal time-series reasoning, the current results do not yet convincingly show that the benchmark is measuring true cross-modal fusion ability. The gains from tool augmentation are also small and unstable, and the current contamination analysis remains weak for numeric time-series data. These limitations should be discussed more explicitly, and the corresponding claims should be narrowed accordingly.

**Strengths And Weaknesses:**

Strengths:
1.The 15-task taxonomy is an excellent contribution. By moving beyond traditional forecasting to include etiological reasoning, abductive reasoning, and decision-making, the authors provide a much-needed holistic framework for time series evaluation.
2.Modality Analysis (Originality): The detailed analysis comparing the intersection and union of Text (T) and Vision (V) modalities offers valuable insights into cross-modal redundancy and the current limitations of modality fusion.
3.High-Quality Error Diagnosis (Presentation): The fine-grained error taxonomy presented in the appendix (differentiating between reasoning, domain knowledge, and perception errors) provides excellent clarity on model failure modes.

Weaknesses
1.The paper reformulates time-series forecasting as a multiple-choice task. While this standardizes the evaluation interface across model families, it also substantially changes the nature of forecasting: models are no longer required to perform open-ended numerical extrapolation, but instead choose among candidate future trajectories. As a result, this setup weakens the benchmark’s ability to evaluate core forecasting challenges. Consistent with this concern, Table 3 shows that Prediction does not scale positively with model size and is, in fact, negatively correlated for both LLMs and VLMs (LLM: -0.2415; VLM: -0.2612), in clear contrast to the positive scaling trends observed for the other dimensions. Therefore, conclusions drawn from this dimension should be stated more cautiously.
2. With 4.1K questions in total, TSRBench is better suited for broad taxonomy coverage than for highly stable comparisons based on small score gaps. The paper also does not report confidence intervals making the robustness of these fine-grained differences unclear.
3. Table 4 shows overall improvements of only 0.5–1.2 points, with large variation across tasks and even degradations on several dimensions. As a result, the conclusions about tool augmentation remain preliminary.

---

> ### Author Rebuttal · Authors · 2026-03-30
>
> We sincerely thank you for the positive feedback and the recognition of our taxonomy as an **excellent contribution** and **valuable insights** to cross-modal redundancy and fusion. For your concerns, we detail the response as follows:
>
> - Q1: Conclusion consistency between MCQ-formatted time series forecasting and open-ended numerical forecasting.
>
> A1: We conduct experiments with LLMs and VLMs on time series forecasting tasks by open-ended numerical forecasting, and use nMAE as the metric. After obtaining the results, we compute the Spearman's rank correlation between LLM and VLM performance and model size, then compare with the MCQ version. The results shown below:
>
> |Spearman $\rho$|MCQ|open-ended forecasting|
> |-|-|-|
> |LLMs|-0.55|-0.61|
> |VLMs|0.03|-0.05|
>
> The results show that the multi-choice format and open-ended numerical forecasting show high correlation: Open-ended forecasting still shows a negative or orthogonal relation to the model size. This indicates that the multiple-choice accuracy is consistent with actual forecasting quality.
>
> - Q2: Confidence intervals for fine-grained comparisons.
>
> A2: We follow your suggestion to report error bars (mean±std, 3 runs). Due to the resource limit during rebuttal, we analyze the performance of 3 LLMs and 3 VLMs. The detailed performance is shown below:
>
> |Model|PR|NU|AD|SA|ER|CD|AR|TR|NR|DR|IR|TSF|EP|QualDM|QuantDM|Overall|
> |-|-|-|-|-|-|-|-|-|-|-|-|-|-|-|-|-|
> |LLMs|||||||||||||||||
> |Qwen2.5-7B|50.7±0.2|50.3±2.7|41.1±4.1|56.0±2.4|14.9±0.3|33.1±1.6|45.4±2.9|27.3±0.8|33.5±0.6|23.1±1.1|27.9±0.9|33.3±0.7|36.3±1.4|29.8±1.3|26.3±1.7|33.2±0.5|
> |Gemma3-12B|49.9±1.5|61.0±1.3|47.5±1.7|65.1±1.2|20.5±1.8|34.6±2.0|43.0±1.3|23.7±1.6|33.2±0.3|28.2±0.5|41.7±2.0|38.7±0.8|33.8±0.6|33.5±0.8|31.3±1.8|36.4±0.8|
> |InternLM3-8B|52.3±1.4|44.0±2.7|36.2±2.1|53.7±1.8|23.0±0.4|25.4±0.7|49.6±2.1|23.5±0.7|31.0±1.2|27.3±2.5|32.7±1.5|36.4±0.6|66.1±0.3|33.6±0.6|25.0±1.5|36.7±0.4|
> |VLMs|||||||||||||||||
> |Qwen2.5-VL-7B|49.8±2.4|55.0±0.4|42.7±0.7|58.9±2.2|21.0±0.8|25.7±1.1|57.3±1.9|27.4±1.1|30.6±0.9|27.9±1.7|30.9±2.4|31.0±0.3|47.9±2.0|32.2±1.5|26.0±0.5|34.7±0.4|
> |InternVL3.5-8B|61.1±1.1|52.3±2.6|51.6±2.0|64.2±2.2|26.1±0.4|38.0±1.0|61.5±1.9|26.5±0.9|40.5±0.4|32.3±0.8|27.3±1.7|39.0±1.1|42.9±1.1|32.0±1.3|22.7±0.5|39.3±0.3|
> |MiMo-VL-7B|59.4±1.3|65.2±2.2|54.3±5.0|66.8±1.2|23.4±0.8|34.5±1.5|67.0±2.1|30.2±0.4|32.9±1.2|25.3±1.3|35.1±1.4|34.6±0.5|29.0±1.3|28.8±0.4|33.8±0.7|37.0±0.2|
>
> The low standard errors (mostly <2%) confirm that performance differences in Table 2 are stable and robust.
>
> - Q3: Scope of tool augmentation.
>
> A3: We observe that time series reasoning involves two qualitatively distinct capability modes that respond oppositely to augmentation:
> (1) For the tasks that require more numerical features (e.g., time series forecasting), the incorporation of tool analysis mitigates the weakness of models on numerical perception and shows consistent improvements.
> (2) For the tasks that need global perception (e.g., Pattern Recognition), the statistics from tools interfered with the model's judgment of the overall pattern, leading to consistent performance degradation. This characterization delimits the practical scope: tool augmentation helps when fine-grained numerical features are the bottleneck, but may be counterproductive when global pattern recognition is required. We will include this discussion in the revision.
>
> - Q4: Data contamination analysis for numeric time-series data.
>
> A4: Our contamination analysis (Appendix D.4) covers numeric time-series data, as we treat them as textual sequences and include them jointly in the n-gram detection pipeline. We further assess on pure nemric time series data following [1]:
>
> |Model|N-gram Accuracy (%)|
> |-|-|
> |o4-mini|0.0%|
> |GPT-5-mini|0.0%|
> |GPT-5|0.1%|
>
> The results show that the numeric time series contamination is minimal.
>
> - Q5: Clarification for cross-modal fusion measurement.
>
> A5: Thank you for your insightful question. Our benchmark primarily focuses on evaluating the capabilities of models on understanding both textual context and time series to conduct reasoning, we agree that the measurement of cross-modal is promising and we suggest two future benchmark design directions: (1) Reciprocal verification tasks that require models to use one representation to verify conclusions drawn from the other (e.g., confirming a visually identified pattern against precise textual values), making cross-modal interaction functionally necessary. (2) Fusion-aware process metrics that go beyond final-answer accuracy to diagnose whether reasoning chains genuinely reference both representations. We will include this future direction discussion in the revision.
>
> References:
>
> [1] Xu, Ruijie, et al. "Benchmarking benchmark leakage in large language models." arXiv preprint arXiv:2404.18824 (2024).

---

> > ### Author Rebuttal · Reviewer_Cgj8 · 2026-04-02
> >
> > Thank you for the detailed rebuttal. My concerns have been addressed, and I will maintain my score.

---

### Official Review · Reviewer_j5is · 2026-03-15

**Soundness:** 2
**Presentation:** 2
**Significance:** 4
**Originality:** 3
**Overall Recommendation:** 3
**Confidence:** 4

**Summary:**

This paper introduces TSRBENCH, a benchmark designed to evaluate the time-series reasoning capabilities of generalist AI models. The benchmark contains 4,125 problems across 14 domains and covers 15 tasks organized into four capability dimensions: perception, reasoning, prediction, and decision-making. It supports multiple input modalities, including textual time-series values, visualized plots, and combined representations, enabling evaluation of both language and multimodal models.

The authors construct the dataset using a mixture of curated real-world data and synthetic simulations, with manually designed questions and verified answers to ensure alignment between time-series signals and contextual information. They then evaluate over 30 proprietary and open-source models, including LLMs, vision-language models, and time-series–specific models, using a unified evaluation protocol.

Experimental results reveal several insights about current models: they perform relatively well on perception tasks but struggle with more complex reasoning, forecasting, and decision-making problems; performance generally scales with model size except for prediction tasks; and textual and visual representations of time series provide complementary information, although current models fail to effectively fuse them. Overall, the benchmark highlights key limitations of existing models and provides a standardized platform for studying time-series reasoning in generalist AI systems.

**Compliance With Llm Reviewing Policy:**

Affirmed.

**Key Questions For Authors:**

How is task difficulty defined and measured across the benchmark? For example, do the authors provide any formal categorization (e.g., easy/medium/hard) or difficulty metrics to ensure balanced evaluation across the 15 tasks?

 Given the large number of domains and tasks, how do the authors quantify diversity at the data level and ensure that tasks are not redundant or highly overlapping across domains?

While some of the time-series data sources are real-world, are the constructed QA-style questions themselves reflective of real decision-making scenarios? Clarifying whether these questions correspond to real analytical workflows would help justify the benchmark design.

Many of the tasks could potentially be solved using traditional statistical or time-series analysis tools. What are the advantages of framing these problems as long-form QA tasks rather than standard numerical analysis tasks?

Could the authors provide more detailed analysis of model performance on synthetic versus real-world tasks, and whether the benchmark results are sensitive to this distinction?

The paper finds that textual and visual time-series representations are complementary but not effectively fused by current models. Do the authors have insights on benchmark design choices that could better evaluate or encourage cross-modal reasoning?

**Limitations:**

yes

**Strengths And Weaknesses:**

Strengths

1. Comprehensive benchmark design: The paper introduces a large-scale benchmark covering 4,125 problems across 14 domains and 15 tasks, providing a broad evaluation of time-series perception, reasoning, prediction, and decision-making.

2. Multi-modal evaluation setup: The benchmark supports textual, visual, and combined time-series inputs, enabling systematic evaluation of both LLMs and multimodal models.

3. Extensive empirical study: The authors evaluate more than 30 proprietary and open-source models and provide several analyses that reveal limitations of current models in time-series reasoning.

Weaknesses

1. Limited benchmark scale: Although diverse, the dataset size is relatively modest compared with many modern reasoning benchmarks.

2. Task difficulty variability: Some tasks rely on multiple-choice formulations and synthetic data, which may not fully reflect real-world time-series reasoning scenarios.

3. Limited analysis of benchmark robustness: The paper provides limited discussion on dataset biases, task difficulty calibration, or potential benchmark leakage beyond basic checks.

---

> ### Author Rebuttal · Authors · 2026-03-30
>
> Thanks for your thoughtful comments.
> - Q1: Benchmark size comparison?
>
> A1 We compare TSRBench with several widely used reasoning benchmarks on size:
> |Benchmark|#Size|
> |-|-|
> |MATH-V[1]|3040|
> |GPQA[2]|448|
> |EMMA[3]|2,788|
> |ROVER[4]|1,312|
> |TSRBench|4125|
>
> TSRBench has a comparable or larger size than others. Our consistent findings further indicate size sufficiency
> - Q2: Real-World Scenario of the MCQ Format?
>
> A2: MCQ format serves as a standardized interface enabling consistent cross-task comparison. Our tasks are grounded in real-world workflows. For example, Qualitative Decision-Making uses real ECG data requiring models to interpret 12-lead signals and select appropriate clinical decisions, mirroring cardiologists' diagnostic workflow
> - Q3: Discussion on dataset biases, task difficulty calibration, and potential benchmark leakage
>
> A3: As shown in A4 and A5 below, TSRBench shows broad difficulty coverage, low task overlap, and high time series diversity, showing minimal dataset biases. For leakage, we conducted both proactive and post-hoc detection (Appendix D), showing minimal data leakage
> - Q4: Task difficulty definition
>
> A4: Following [5], we apply Item Response Theory (Rasch model) across models to estimate a continuous difficulty $b$ for each question, where a higher $b$ indicates greater task difficulty:
> |Task|$b_{mean}$|$b_{std}$|
> |-|-|-|
> |AD|0.89|1.73|
> |SA|0.31|1.95|
> |NU|0.67|1.94|
> |PR|0.70|1.80|
> |CD|1.53|0.97|
> |AR|−0.37|0.79|
> |TR|1.42|1.02|
> |NR|1.39|1.32|
> |DR|1.43|0.72|
> |IR|1.53|1.07|
> |TSF|1.60|1.03|
> |EP|0.16|0.87|
> |QualDM|1.48|1.85|
> |QuantDM|1.95|1.42|
> |Overall|1.22|1.42|
>
> The results show (1) every task has $b_{std}>0.7$, ensuring within-task discrimination; (2) tasks span from AR ($b_{mean}$=−0.37) to QuantDM (1.95), providing a clear difficulty gradient among tasks; (3) overall positive skew ($b_{mean}$=1.22) ensures the difficulty for future models
> - Q5: Time series diversity quantification and task non-redundancy
>
> A5: (1) Following GIFT-Eval [6], we compute six statistical features across all domains and report the per-domain mean:
> |Domain|trend|linearity|entropy|hurst|stability|lumpiness|
> |-|-|-|-|-|-|-|
> |Finance|0.5|2.27|0.42|0.96|0.85|0.11|
> |Climate|0.16|0.06|0.53|0.88|0.33|0.3|
> |Traffic|0.44|0.2|0.54|0.86|0.71|0.05|
> |Energy|0.25|0.11|0.48|0.84|0.62|0.15|
> |Unlabeled|0.71|1.65|0.52|0.81|0.62|0.2|
> |Health|0.33|0.99|0.62|0.63|0.51|1.49|
> |Meteorology|0.35|0.08|0.57|0.79|0.19|0.18|
> |Hydrology|0.51|1.59|0.5|0.93|0.83|1.18|
> |Physics|1.0|-6.67|0.27|0.98|1.02|0.02|
> |Astronomy|0.5|1.16|0.65|0.76|0.46|0.62|
> |Agriculture|0.11|1.02|0.71|0.71|0.3|0.18|
> |Political Science|0.16|0.09|0.89|0.65|0.14|0.32|
> |Seismology|0.04|0.16|0.9|0.57|0.1|0.22|
> |Chemistry|0.03|0.05|0.59|0.93|0.93|0.5|
>
> The statistics reveal distinctly different profiles, confirming high time-series diversity
>
> (2) Task Diversity: We compute the Spearman rank correlation matrix of accuracy vectors across models. The correlation results (https://imgbox.com/MBFWBCXS) show that all task-pair correlations $|ρ|<0.5$, confirming low task redundancy
> - Q6: Advantages over traditional statistical analysis
>
> A6: Most tasks cannot be solved by traditional tools alone, as they require understanding context for the final conclusion. This is where TSRBench differs from traditional time series analysis (§3.6), requiring the model to use both textual context and numericaltime series for complex reasoning, rather than simply numerical prediction
> - Q7: Synthetic vs. real-world task sensitivity
>
> A7: We compute average accuracy on synthetic data tasks (NR, DR, QuantDM) vs. real-world data tasks (remaining 12 tasks) across all models:
> |Synthetic|Real-world|
> |-|-|
> |35.8%|46.3%|
>
> Synthetic tasks are more challenging due to precise numerical requirements. We further compute the Spearman rank correlation between model rankings on the two splits, obtaining $\rho=0.92$. The strong correlation confirms that model rankings are highly consistent across both data types, confirming that our findings are robust
> - Q8: Insights on Benchmark Design for Cross-modal Reasoning
>
> A8: To better evaluate cross-modal reasoning, we suggest two directions: (1) Reciprocal verification tasks requiring models to use one modality to verify conclusions from the other, making cross-modal interaction necessary (2) Evaluation metrics assess whether reasoning chains ground both modalities rather than a single modality. We will include this discussion in the revision.
>
> [1] Measuring multimodal mathematical reasoning with math-vision dataset, Neurips 2024
>
> [2] Gpqa: A graduate-level google-proof q&a benchmark, First conference on language modeling, 2024
>
> [3] Can MLLMs Reason in Multimodality? EMMA: An Enhanced MultiModal ReAsoning Benchmark, ICML 2025
>
> [4] ROVER: Benchmarking Reciprocal Cross-Modal Reasoning for Omnimodal Generation, ICLR 2026
>
> [5] Redundancy principles for mllms benchmarks, ACL 2025
>
> [6] Gift-eval: A benchmark for general time series forecasting model evaluation, Arxiv 2024

---

### Decision · Program_Chairs · 2026-04-30

**Decision:**

Accept (regular)

**Comment:**

This paper presents TSRBench, a comprehensive multi-task, multi-modal benchmark for evaluating generalist models on time-series understanding, which covers four capability dimensions: perception, reasoning, prediction, and decision-making. Across a large-scale evaluation of more than 30 language, vision-language, and time-series models, the paper provides a testbed that surfaces key limitations of current systems, especially in forecasting and cross-modal reasoning. Reviewers found the benchmark timely and valuable. In particular, this new benchmark includes a well-motivated taxonomy with 15 tasks, and it combines both real-world and synthetic data with human verification. In addition, it supports multiple input modalities. Meanwhile, reviewers raised a few concerns, including the use of multiple-choice reformulations for prediction tasks, the lack of detailed robustness analyses, etc. These concerns were mostly addressed by the authors’ rebuttal. Overall, the paper makes a meaningful benchmark contribution to the community, and I encourage the authors to incorporate the additional clarifications and any new results discussed in the rebuttal into the final version.